# MIRACLE: Causally-Aware Imputation via Learning Missing Data Mechanisms

**Trent Kyono**\*
University of California, Los Angeles
tmkyono@ucla.edu

**Yao Zhang**\*
University of Cambridge
yz555@cam.ac.uk

**Alexis Bellot**
University of Oxford
Oxford, United Kingdom
alexis.bellot@eng.ox.ac.uk

**Mihaela van der Schaar**
University of Cambridge
University of California, Los Angeles
The Alan Turing Institute
mv472@cam.ac.uk

## Abstract

Missing data is an important problem in machine learning practice. Starting from the premise that imputation methods should preserve the causal structure of the data, we develop a regularization scheme that encourages any baseline imputation method to be causally consistent with the underlying data generating mechanism. Our proposal is a causally-aware imputation algorithm (MIRACLE). MIRACLE iteratively refines the imputation of a baseline by simultaneously modeling the missingness generating mechanism, encouraging imputation to be consistent with the causal structure of the data. We conduct extensive experiments on synthetic and a variety of publicly available datasets to show that MIRACLE is able to consistently improve imputation over a variety of benchmark methods across all three missingness scenarios: at random, completely at random, and not at random.

## 1 Introduction

Missing data is an unavoidable byproduct of collecting data in most practical domains. In medicine, for example, doctors may choose to omit what they deem to be irrelevant information (e.g., some patients may be asked to get comprehensive blood tests while others don't), data may be explicitly omitted by the patient (e.g., avoiding questions on smoking status precisely because of their smoking habit) or simply misrecorded in electronic health systems (see e.g., [3, 12, 28]).

Imputation algorithms can be used to estimate missing values based on data that was recorded, but their correctness depends on the type of missingness. For instance, expanding on the example above, younger patients may also be more likely to omit their smoking status. As illustrated in Figure 1, the challenge is that implicitly conditioning inference on observed data introduces a spurious path of correlation between age and the prevalence of smoking that wouldn't exist with complete data.

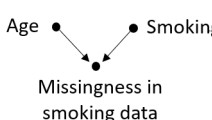

Figure 1: Missingness may introduce spurious dependencies.

Missing data creates a shift between the available missing data distribution and the target complete data distribution. It is a shift that may be explicitly modeled as missingness indicators in an underlying causal model (i.e., a missingness graph as proposed by Mohan et al. [19]) as shown in Figure 1. The learning problem is one of *extrapolation*, learning with access to a missing data distribution for

---

\*Equal contribution

35th Conference on Neural Information Processing Systems (NeurIPS 2021).

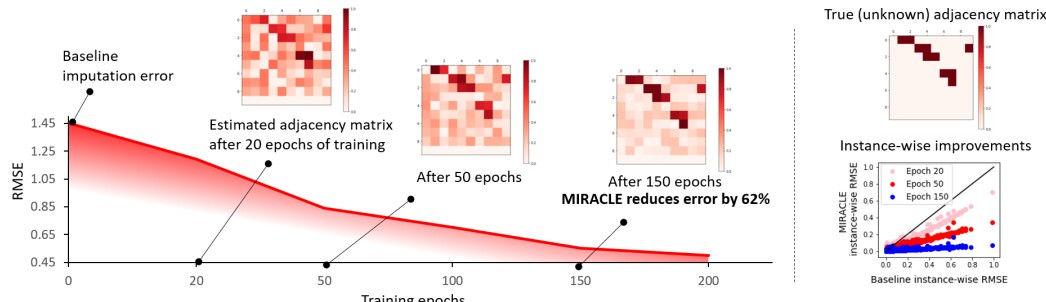

Figure 2: MIRACLE refines baseline imputation by simultaneously learning an $m$-graph using a bootstrap imputation loop that serves to incrementally regularize predictions with a learned causal graph. We plot average testing error and estimated causal graph as a function of training epochs on a synthetic data experiment described in Section 4. The true causal structure (as an adjacency matrix) and imputation improvements for each missing value separately (each missing value with a corresponding dot) is shown in the right-most panel.

prediction and inference on the complete data distribution – that is, generated from a model where all missingness indicators have been intervened on (interventions interpreted in the sense of Pearl [22]) thus graphically removing the dependence between missingness and its causes, and any spurious correlations among its ancestors.

With this causal interpretation, imputation of missing data on a given variable $Y$ from other observed variables $X$ is formulated as a problem of robust optimization,

$$\underset{\theta \in \Theta}{\text{minimize}} \sup_{P \in \mathcal{P}} \mathbb{E}_{(X,Y) \sim P} \left[ (f_\theta(X) - Y)^2 \right], \tag{1}$$

simultaneously optimizing over the set of distributions $\mathcal{P}$ arising from interventions on missingness indicators. Causal solutions – i.e. imputation using functions of causal parents of each missing variable in the underlying causal graph – are a closely-related version of this problem with an uncertainty set $\mathcal{Q}$ defined as any distribution arising from interventions on observed variables and variable indicators (see e.g. sections 3.2 and 3.3 in [16]),

$$\sup_{P \in \mathcal{P}} \mathbb{E}_{(X,Y) \sim P} \left[ (f_\theta(X) - Y)^2 \right] \le \sup_{P \in \mathcal{Q}} \mathbb{E}_{(X,Y) \sim P} \left[ (f_\theta(X) - Y)^2 \right], \tag{2}$$

since $\mathcal{P} \subset \mathcal{Q}$. Our premise is that causal solutions, i.e. minimizing the right-hand-side of (2), are expected to correct for spurious correlations introduced by distribution shift due to missing data and preserve the dependencies of the complete data for downstream analysis.

## 1.1 Contributions

In this paper, we propose to impute while preserving the causal structure of the data. Missing values in a given variable are replaced with their conditional expectation given the realization of its causal parents instead of the more common conditional expectation given all other observed variables, which absorbs spurious correlations.

We propose a novel imputation method called Missing data Imputation Refinement And Causal LEarning (MIRACLE). MIRACLE is a general framework for imputation that operates on any baseline (existing) imputation method. A visual description is given in Figure 2: given some initial imputation from a baseline method, MIRACLE refines its imputations iteratively by learning a missingness graph ($m$-graph) [19] and regularizing the imputation function such that it is consistent with the causal graph generating the data, substantially improving performance. In experiments, we apply MIRACLE to improve six popular imputation methods as baselines. We present detailed simulations to demonstrate on synthetic and a variety of publicly available datasets from the UCI Machine Learning Repository [8] that MIRACLE can improve imputation in almost every scenario and never degrades performance across all imputation methods.

## 1.2 Related work

The literature on imputation is large and varied. Still, most imputation algorithms work with the prior assumption that the missing data mechanism is ignorable, in the sense that imputation does

not require explicitly modeling the distribution of missing values for imputation [25]. Accordingly, classical imputation methods impute using a joint distribution over the incomplete data that is either explicit or implicitly defined through a set of conditional distributions. For example, explicit joint modeling methods include generative methods based on Generative Adversarial Networks [33, 34, 1], matrix completion methods [15], and parametric models for the joint distribution of the data. Missing values are then imputed by drawing from their predictive distribution. The conditional modeling approach [31] consists of specifying one model for each variable and iteratively imputing using estimated conditional distributions. Examples of discriminative methods are random forests [27], autoencoders [11, 10, 14], graph neural networks [35], distribution matching via optimal transport [20], and multiple imputation using chained equations [6].

In a different line of research, Mohan et al., in a series of papers, see e.g. [19, 18], explicitly considered missing data within the underlying causal mechanisms of the data. Subsequently, a range of related problems has been studied, including identifiability of distributions and causal effects in the presence of missing data, see e.g. [5, 26, 21], testable implications relating to the causal structure using missing data [18], and causal discovery in the presence of missing data [9, 30]. Our focus, in contrast, is algorithmic in nature. We aim to develop an algorithm that improves imputation quality by leveraging causal insights represented as an estimated missingness graph learned from data.

## 2 Background

The basic semantic framework of our analysis rests on structural causal models (SCMs) (see e.g. Chapter 7 in [22] for more details) explicitly introducing missingness indicators and their functional relationship with other variables, using in part the notation of [19]. We define an SCM $\mathcal{M}$ as a tuple $(\boldsymbol{X}, \boldsymbol{R}, \boldsymbol{U}, \mathcal{F}, P)$ where $\boldsymbol{X}$ is a vector of $d$ endogenous variables and $\boldsymbol{U}$ is a vector of exogenous variables.[2] $\boldsymbol{R}$ is the vector of missingness indicators that represent the status of missingness of the endogenous variables $\boldsymbol{X}$. Precisely, $R_j$ is responsible for the value of a proxy variable $Z_j$ of $X_j$, i.e., the observed version of $X_j$. For example, $Z_j$ is equal to $X_j$ if the corresponding record is observed ($R_j = 1$), otherwise $Z_j$ is missing ($R_j = 0$). $\mathcal{F}$ is a set of functions where each $f_X, f_R \in \mathcal{F}$ decide the values of an endogenous variable $X$ and a missingness indicator variable $R$, respectively. The function $f_X$ takes two separate arguments as parts of $\boldsymbol{X}$ (except $X$ itself) and $\boldsymbol{U}$, termed as $\mathrm{Pa}_X$ and $U_X$. That is, $X \leftarrow f_X(\mathrm{Pa}_X, U_X)$ and $R \leftarrow f_R(\mathrm{Pa}_R, U_X)$.

The randomness in SCMs comes from the exogenous distribution $P_{\boldsymbol{U}}(\boldsymbol{u})$ where the exogenous variables in $\boldsymbol{U}$ are generated independently and are mutually independent. Naturally, through the functions in $\mathcal{F}$, the SCM $\mathcal{M}$ induces a joint distribution $P_{\boldsymbol{X}}(\boldsymbol{x})$ over the endogenous variables $\boldsymbol{X}$, called the endogenous distribution. An intervention on some arbitrary random variables $\boldsymbol{V}$ in $\boldsymbol{X}$ and $\boldsymbol{R}$, denoted by $do(\boldsymbol{v})$, is an operation which sets the value of $\boldsymbol{V}$ to be $\boldsymbol{v}$, regardless of how they are ordinarily determined. For an SCM $\mathcal{M}$, let $\mathcal{M}_{\boldsymbol{v}}$ denote a submodel of $\mathcal{M}$ induced by intervention $do(\boldsymbol{v})$. The interventional distribution $P_{\boldsymbol{X}}(\boldsymbol{x}|do(\boldsymbol{v}))$ induced by $do(\boldsymbol{v})$ is defined as the distribution over $\boldsymbol{X}$ in the submodel $\mathcal{M}_{\boldsymbol{v}}$, namely, $P_{\boldsymbol{X}, \mathcal{M}_{\boldsymbol{v}}}(\boldsymbol{x}) = P_{\boldsymbol{X}}(\boldsymbol{x}|do(\boldsymbol{v}))$.

Each SCM in the context of missingness is associated with a $m$-graph $\mathcal{G}$ (e.g., Fig. 1a), which is a directed acyclic graph (DAG) where nodes represent endogenous variables $\boldsymbol{X}$ and missingness indicators $\boldsymbol{R}$, and arrows represent the arguments $\mathrm{Pa}_X$ and $\mathrm{Pa}_R$ of each function $f_X$ and $f_R$ respectively. By convention, exogenous variables $\boldsymbol{U}$ are often not shown explicitly in the graph.

**Assumption 1 (Missingness indicators are not causes)** *No missingness indicator in $\boldsymbol{R}$ can be the cause of the endogenous variables $\boldsymbol{X}$, i.e., the arguments of the functions generating $\boldsymbol{X}$.*

**Assumption 2 (Causal sufficiency)** *Exogeneous variables $\boldsymbol{U}$ are mutually independent, i.e., all common parents of the endogenous variables are included in $\boldsymbol{X}$.*

**Assumption 3 (No self-masking missingness)** *Self-masking missingness refers to missingness in a variable that is caused by itself. In the $m$-graph this is depicted by an edge from $X_j$ to $R_j$ (as shown in Figure 3 (d)). We assume that there is no such edges in the $m$-graph.*

**Assumption 4 (Observed root nodes)** *The endogenous variables $X_j$ such that $Pa_{X_j} = \emptyset$ (i.e., the root nodes) in the $m$-graph are always observed ($R_j = 1$ with probability 1).*

---

[2]Essentially, $\boldsymbol{X}$ is the ground-truth features; $\boldsymbol{U}$ is the random noise in the data generating process.

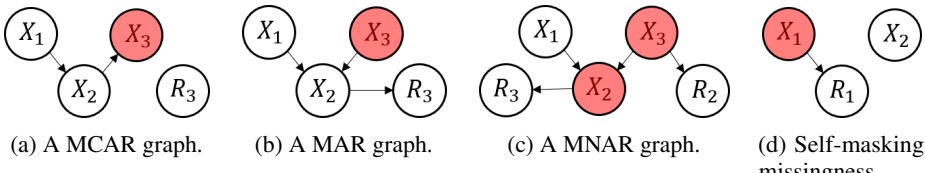

| (a) A MCAR graph. | (b) A MAR graph. | (c) A MNAR graph. | (d) Self-masking missingness. |

Figure 3: Example graphs. $\boldsymbol{X} = (X_1, X_2, X_3)$ are endogenous variables and $\boldsymbol{R} = (R_1, R_2, R_3)$ are missing data indicators. Red shaded variables are not always observed while white shaded variables are always observed.

We make the four assumptions above throughout the following sections. Assumption 1 and 2 are employed in most related works using $m$-graphs (see e.g. [18, 19]). Assumption 1 is valid, for example, if $\boldsymbol{R}$ is generated in the data collection process after the variable values are assigned. Consequently, under this assumption, if two endogenous variables of interest $X_1$ and $X_2$ are not $d$-separated by some variable $X_3$, they are not $d$-separated by $X_3$ together with their missingness indicators $R_1$ and $R_2$. We denote an independent relation in a data distribution by "$\perp\!\!\!\perp$" and $d$-separation in a $m$-graph by "$\perp\!\!\!\perp_d$". We assume the data distribution is faithful to a $m$-graph, meaning that the two independencies are equivalent. As shown in Figure 3, data is missing completely at random (MCAR) if $\boldsymbol{X} \perp\!\!\!\perp_d \boldsymbol{R}$ holds in the $m$-graph, missing at random (MAR) if for any endogenous variable $X_j$, $R_j \perp\!\!\!\perp_d X_j \mid \boldsymbol{X}_{-j}$ holds, and missing not at random (MNAR) otherwise, as stated in [19]. If Assumption 3 is violated, we are unable to learn the missingness for self-masked variables. Assumption 4 is necessary for imputing all the missing variables from their causal parents. These assumptions are imperative for MIRACLE to provide improved imputations by leveraging the causal structure of the underlying data generating process. In our experiments (Section 4), we apply MIRACLE to real-world datasets where these assumptions are not guaranteed.

## 2.1 Why is imputation prone to bias?

The reason for considering the causal structure of the underlying system is that when learning an imputation model from observed data, implicitly conditioning on some missingness indicators in $\boldsymbol{R}$ induces spurious dependencies that would not otherwise exist. For example, in a graph $X_1 \rightarrow R_3 \leftarrow X_2$, conditioning on $R_3 = 1$ induces a dependence between $X_1$ and $X_2$. In general, the distributions $P_{\boldsymbol{X}}(\boldsymbol{x} | \boldsymbol{R} = \boldsymbol{r})$ and $P_{\boldsymbol{X}}(\boldsymbol{x} | do(\boldsymbol{r}))$ differ unless missingness occurs completely at random, and motivates an interpretation of the problem as domain generalization, training on data from one distribution ultimately to be applied on data from a different distribution that, in our case, arises from missing data (i.e., interventions on missingness indicators). This shift is not addressed in the imputation methods that only use the feature correlations.

## 3 MIRACLE

In this section, we propose to correct for the shift in distribution due to missing data by searching for causal solutions and explicitly refining imputation methods using a penalty on the induced causal structure. In practice, we have $n$ $i.i.d.$ realizations of the observed version of $\boldsymbol{X} \in \mathbb{R}^d$, concatenated as a incomplete data matrix $\mathbf{X} \in \mathbb{R}^{n \times d}$, together with missingness indicators concatenated in a matrix $\mathbf{R} \in \{0, 1\}^{n \times d}$. We use here the same bold uppercase notation for sets of variables and matrices of observations but their meaning should be clear from the context. Our goal is to impute the unobserved values in $\mathbf{X}$ using each variable's causal parents. We define the *imputed* data $\tilde{\mathbf{X}} \in \mathbb{R}^{n \times d}$,

$$\tilde{\mathbf{X}} = \mathbf{R} \odot \mathbf{X} + (1 - \mathbf{R}) \odot \hat{\mathbf{X}}$$

where $\odot$ is the element-wise product of matrices and $\hat{\mathbf{X}}$ is an estimate of the complete data matrix.

## 3.1 Network architecture

In this section, we describe our approach for estimating $\hat{\mathbf{X}}$. Let $d_S \leq d$ be the number of partially observed features, i.e., missing for at least one realization. $S$ is the set of missing features indices. The imputation network is defined as a function $f : \mathbb{R}^d \rightarrow \mathbb{R}^d \times [0, 1]^{d_S}$ that takes an initially imputed dataset $\tilde{\mathbf{X}}^{(0)}$ (using an existing baseline imputation method), and returns two quantities:

1. A refined imputation $\hat{\mathbf{X}}$.

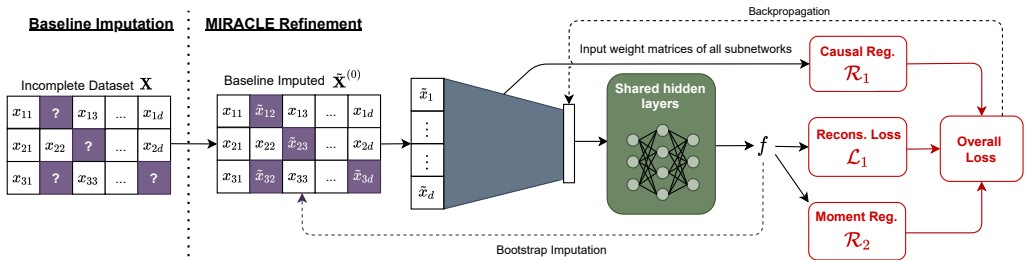

Figure 4: Network and optimization diagram for MIRACLE.

2. An estimation of the probabilities of features $X_{ij}$ being missing, $i = 1..., n$ and $j \in S$.

A depiction of the network architecture and optimization algorithm is shown in Figure 4. The architecture is constructed with respect to the assumptions shown in Section 2. Our model $f$ is decomposed into two sub-networks, $f = (f^{(imp)}, f^{(miss)})$, responsible for imputing the unobserved data and estimate the probabilities of missingness, respectively. The imputation network has $d$ components, $f^{(imp)} = (f_1^{(imp)}, \ldots, f_d^{(imp)})$, one for each variable, and the missingness network has $d_S$ components, $f^{(miss)} = (f_1^{(miss)}, \ldots, f_{d_S}^{(miss)})$. Each component, for both networks, has separate input and output layers but shared hidden layers (of size $h$). Let $\mathbf{W}_{1,j}^{(imp)}$ and $\mathbf{W}_{1,j}^{(miss)}$ denote the $h \times d$ weight matrix (we omit biases for clarity) in the input layer of $f_j^{(imp)}$ and $f_j^{(miss)}$ respectively. The $j$-th column of $\mathbf{W}_{1,j}^{(imp)}$ and $\mathbf{W}_{1,j}^{(miss)}$ is set to $\mathbf{0}$. Let $\mathbf{W}_m \in \mathbb{R}^{h \times h}$, for $m = 2, \ldots, M - 1$, denote the weight matrix of each hidden layer and let $\mathbf{W}_{M,j}^{(imp)}$ and $\mathbf{W}_{M,j}^{(miss)}$, be the $1 \times h$ dimensional output layers of each sub-network. The imputation network prediction is given by,

$$f_j^{(imp)}(\mathbf{x}) := \mathbf{W}_{M,j}^{(imp)} \phi(\cdots \phi(\mathbf{W}_2 \phi(\mathbf{W}_{1,j}^{(imp)} \mathbf{x}))),$$

for $j = 1, \ldots, d$. And similarly, the missingness network prediction is given by,

$$f_j^{(miss)}(\mathbf{x}) := \sigma\big(\mathbf{W}_{M,j}^{(miss)} \phi(\cdots \phi(\mathbf{W}_2 \phi(\mathbf{W}_{1,j}^{(miss)} \mathbf{x})))\big),$$

for $j = 1, \ldots, d_S$, where $\phi(\cdot)$ is the ELU activation function and $\sigma$ is the sigmoid function. Our network is optimized with respect to three objectives. First, to accurately predict missing values, second, to faithfully encode the causal relationships given by the underlying $m$-graph, and third to satisfy a moment constraint of the missing data mechanism on the imputed values.

## 3.2 Reconstruction loss

The first objective is to train $f$ to correctly reconstruct each feature from the observed data using a reconstruction loss,

$$\mathcal{L}_1 = \frac{1}{n}\left( \sum_{i=1}^{n} \big\| \mathbf{x}_i \odot \mathbf{r}_i - f^{(imp)}(\tilde{\mathbf{x}}_i^{(0)}) \odot \mathbf{r}_i \big\|^2 + \sum_{i=1}^{n} \text{CrossEntropy}\Big[\tilde{\mathbf{r}}_i, f^{(miss)}(\tilde{\mathbf{x}}_i^{(0)})\Big] \right),$$

where $\mathbf{x}_i$ and $\tilde{\mathbf{x}}_i^{(0)}$ are the realized and imputed feature vector of the $i$-th instance, $\tilde{\mathbf{r}}_i$ are the $d_S$ components of $\mathbf{r}_i$ that are missing for at least one instance. The first loss term is for reconstructing the observed features, and the second loss term is for estimating the probabilities of missingness.

## 3.3 Causal regularizer

The second objective is to ensure that the dependencies defined by $f$ correspond to a DAG over the features $X$ and the missing indicators $R_S$, which enforces that the learned functional dependencies recover a DAG in the equivalence class of causal graphs over the observed data. Enforcing the acyclicity of the dependencies induced by a continuous function $f$ is originally proposed in [36, 37]. Define a binary adjacency matrix $\mathbf{B} \in \{0,1\}^{(d+d_S) \times (d+d_S)}$; $[\mathbf{B}]_{k,j} = 0$ (i.e., the $l_2$-norm of the $k$-th column of the matrix $\mathbf{W}_{1,j}^{(imp)}$ or $\mathbf{W}_{1,j}^{(miss)}$ is 0) is a realistic and sufficient condition for achieving $\partial_k f_j = 0$. The adjacency matrix $\mathbf{B}$ of the graph induced by the learned $f$ is acyclic if and only if,

$$\mathcal{R}_1 = \tfrac{1}{2} h^2(\mathbf{B}) + h(\mathbf{B}), \tag{3}$$

is equal to zero, where $h(B) := \mathrm{Tr}(\exp\{\mathbf{B} \odot \mathbf{B}\}) - (d + d_S)$ and $\exp(\cdot)$ is the matrix exponential.

**Remark 1.** Existing imputation methods based on feature correlations essentially assume an undirected (non-causal) graph between the features. Further, acyclicity is a realistic and practical assumption to make on the static datasets collected by human experts. In nature, most data distributions generate their features in some order. In a directed graph, a cycle means a path starts and ends at the same node. This is unlikely to happen in the data generating process if not considering variables over time, i.e., time-series data. By enforcing acyclicity, MIRACLE only uses the causal parents for imputation, which is less biased by spurious dependencies that only exist in the observed data.

## 3.4 Moment regularizer

The third objective leverages a set of moment constraints in the missingness pattern to improve imputation. Assume $\xi_j = P(R_j = 1 \mid \mathrm{Pa}_{R_j})) \in (\delta, 1 - \delta)$, for some $\delta > 0$. The following derivation holds for MAR or MCAR missingness patterns only. It holds that,

$$\mathbb{E}\left\{\frac{R_j X_j}{\xi_j}\right\} = \mathbb{E}\left\{\mathbb{E}\left[\frac{R_j X_j}{\xi_j} \mid X_j, \mathrm{Pa}_{R_j}\right]\right\} = \mathbb{E}\left\{\frac{X_j}{\xi_j}\mathbb{E}\left[R_j \mid X_j, \mathrm{Pa}_{R_j}\right]\right\} = \mathbb{E}\{X_j\}, \quad (4)$$

where the third equality follows from the MAR assumption ($R_j \perp\!\!\!\perp X_j \mid \boldsymbol{X}_{-j}$). Under the MCAR assumption, this derivation holds trivially since in that case $R_j \perp\!\!\!\perp X_j$.

We can use the missingness and imputation networks to enforce the above equality algorithmically, ensuring the left hand side equals the right hand side in the empirical version of (4) as follows,

$$\mathcal{R}_2 = \sum_{j=1}^{d_S} [\hat{\tau}_{j,\mathrm{SIPW}} - \hat{\tau}_{j,\mathrm{mean}}]^2 = \sum_{j=1}^{d_S}\left[\left(\sum_{i=1}^{n} e_{ij}r_{ij}\right)^{-1}\sum_{i=1}^{n} e_{ij}r_{ij}x_{ij} - \frac{1}{n}\sum_{i=1}^{n} f_{S[j]}^{(imp)}(\tilde{\mathbf{x}}_i^{(0)})\right]^2,$$

where $e_{ij} = 1/f_j^{(miss)}(\tilde{\mathbf{x}}_i^{(0)})$, and $S[j]$ is the $j$-th element of $S$, i.e., the index of the $j$-th missing feature. Minimizing $\mathcal{R}_2$ forces the two estimators of $\mathbb{E}\{X_j\}$ to match, the stabilized inverse propensity score weighting (SIPW) estimator $\hat{\tau}_{j,\mathrm{SIPW}}$ [24] using the missingness network $f_j^{(miss)}$ (in $e_{ij}$) and the mean estimator $\hat{\tau}_{j,\mathrm{mean}}$ using the imputation network $f_{S[j]}^{(imp)}$.

**Remark 2.** We hypothesize this mechanism can improve performance for two reasons. First, the missing data mechanism $P(R_j = 1 \mid \mathrm{Pa}_{R_j})$ can be a simpler function that takes less samples to learn than the function that generates the feature $j$, $\mathbb{E}[X_j \mid \mathrm{Pa}_{X_j}]$. Then the SIPW estimator based on $f_j^{(miss)}$ will converge to the true mean faster than the estimator based on $f_{S[j]}^{(imp)}$. Second, in $\mathcal{R}_2$, the mean estimator using $f_{S[j]}^{(imp)}$ is based on all the samples; $f_{S[j]}^{(imp)}$ is trained to produce predictions on the samples with missing feature $j$ for the sake of matching the SIPW estimator. By contrast, without the regularizer $\mathcal{R}_2$, $f_{S[j]}^{(imp)}$ is solely trained on the samples with observed feature $j$, and its performance may fail to generalize to data with missing feature $j$.

## 3.5 Bootstrap Imputation

Discovering a causal graph requires complete data. However, this is not the case for missing data problems. Because of this, we require that MIRACLE be seeded by another imputation method. Imputed values are iteratively refined by MIRACLE, hence "bootstrapping", to potentially converge to a new imputation that minimizes MIRACLE's objective (including causal and moment regularizers). MIRACLE's objective for optimization is,

$$\mathcal{L} = \mathcal{L}_1 + \beta_1\mathcal{R}_1 + \beta_2\mathcal{R}_2, \quad (5)$$

where $\beta_1$ and $\beta_2$ are hyperparameters that define the strength of regularization. We iteratively update the baseline matrix $\tilde{\mathbf{X}}^{(0)}$ with a new imputed matrix $\tilde{\mathbf{X}}$ given by MIRACLE every ten epochs in training. With increasing epochs, stochastic optimization minimizes the loss for the imputed matrices that respect the causal and moment regularization. In theory, this is analogous to supervised training a denoising autoencoder (DAE) [32, 4, 23], but differs only by the fact that "noise" comes from prior or previous imputations. In training DAE, the input samples are corrupted by independent noise

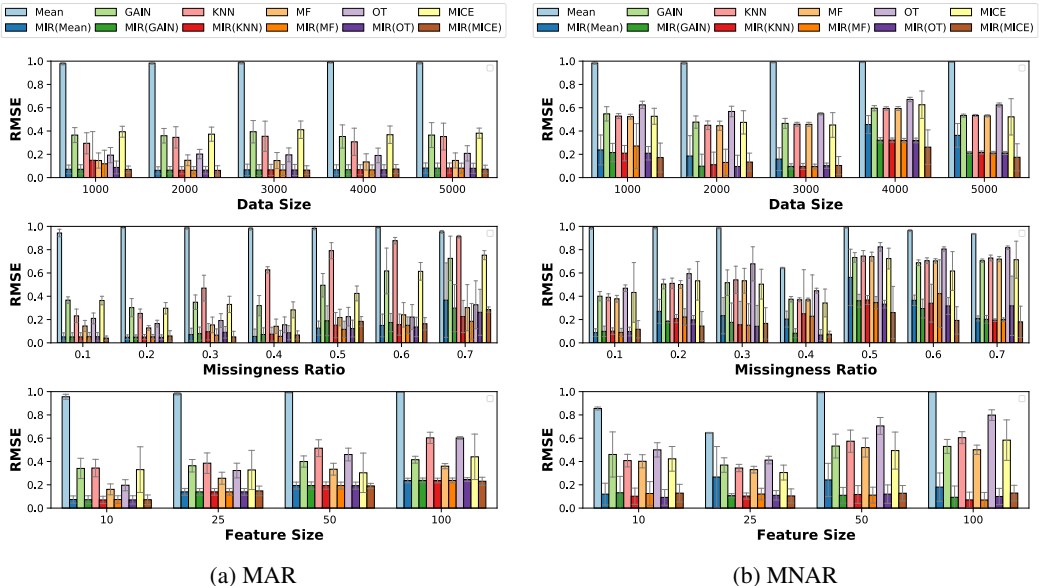

(a) MAR          (b) MNAR

Figure 5: **Experiments on MAR (left) and MNAR (right) synthetic data** in terms of RMSE over varying *dataset sizes* (**top**), *missingness rates* (**middle**), and *feature sizes* (**bottom**). Note that we show the average error over a variety of DAG instantiations and target variables, thus the magnitude and standard deviation of errors vary significantly between runs. See Appendix B for MCAR results.

with each epoch, yet convergence is still guaranteed [2]. In our experiments, we demonstrate that bootstrap imputation indeed converges on multiple datasets and baseline methods. In Appendix E, we provide experiments on the computational complexity of MIRACLE.

## 4    Experiments

In this section, we validate the performance of MIRACLE using both synthetic and a variety of real-world datasets.

1. In the first set of experiments, we quantitatively analyze MIRACLE on synthetic data generated from a known causal structure.

2. In the second set of experiments, we quantitatively evaluate the imputation performance of MIRACLE using various publicly available UCI datasets [8].

**General set-up.** We conduct each experiment five times under random instantiations of missingness. We report the RMSE along with standard deviation across each of the five experiments. Unless otherwise specified, missingness is applied at a rate of 30% per feature. For MCAR, this is applied uniformly across all features. For MAR, we randomly select 30% of the features to have missingness caused by another disjoint and randomly chosen set of features. Similarly, we randomly select 30% of features to be MNAR. We induce MAR and MNAR missingness using the methods outlined in [33], and we provide more details in Appendix A.2.

We use an 80-20 train-test split. We performed a hyperparameter sweep (log-based) for $\beta_1$ and $\beta_2$ with ranges between 1e-3 and 100. By default we have $\beta_1$ and $\beta_2$ set to 0.1 and 1, respectively.

**Evaluating imputation.** For each subsection below, we present three model evaluations in terms of missingness imputation performance, label prediction performance of a prediction algorithm trained on imputed data and the congeniality of imputation models.

- **Missingness imputation performance** is evaluated with the root mean squared error comparing the imputed missing values with their actual unobserved values.

- **Label prediction performance** of an imputation model is its ability to improve the post-imputation prediction. By post-imputation, we refer to using the imputed data to perform a

downstream prediction task. To be fair to all benchmark methods, we use the same model (support vector regression) in all cases.

- **The congeniality** of an imputation model is its ability to impute values that respect the feature-label relationship post imputation. Specifically, we compare, support vector parameters, $w$, learned from the complete dataset with the parameters $\hat{w}$, learned from the imputed dataset. We report root mean square error $(||w - \hat{w}||^2)^{1/2}$ for each method. Lower values imply better congeniality [33].

**Baseline imputation methods.** We apply MIRACLE imputation over a variety of six commonly used imputation baseline methods: (1) mean imputation using the feature-wise mean, (2) a deep generative adversarial network for imputation using GAIN [33] (3) $k$-nearest neighbors (KNN) [29] using the Euclidean distance as a distance metric of each missing sample to observed samples, (4) a tree-based algorithm using MissForest (MF) [27], (5) a deep neural distribution matching method based on optimal transport (OT) [20], and (6) Multivariate Imputation by Chained Equations (MICE) [6]. For each of the baseline imputation methods with tunable hyperparameters, we used the published values. We implement MIRACLE using the `tensorflow`[3] library. Complete training details and hyperparameters are provided in Appendix D.

**Additional experiments.** In Appendix B and C, we also evaluate MIRACLE in terms of providing imputations that improve **predictive performance** [33, 35], as well as, in its ability to impute values that respect the feature-label relationship, i.e., **congeniality** [17, 7]. We also investigate the impact of the graphical missingness location on imputation performance on synthetic data in Appendix G.

## 4.1 Synthetic data

In this subsection, we evaluate MIRACLE on synthetic data. In doing so, we can control aspects of the underlying data generating process and possess oracle knowledge of the DAG structure.

**Data generating process.** We generate random Erdos-Renyi graphs with functional relationships from parent to children nodes. At each node, we add Gaussian noise with mean 0 and variance 1. For a complete description of the underlying data generating process, see Appendix A.

**Synthetic results.** In Figure 5, we show experiments of MIRACLE on synthetic MAR data in terms of RMSE. Our experiments show that MIRACLE is able to significantly improve imputation over each of the baseline imputation methods. Figure 5 shows MIRACLE improves performance over each baseline method across various **dataset sizes, missingness ratios, and feature sizes (DAG sizes)**. We show results for MCAR in Appendix B.

## 4.2 Experiments on real data

In Figure 6 we show experiments of MIRACLE on real data. We perform experiments on several UCI datasets used in [33, 35, 20, 13]: Autism, Life expectancy, Energy, Abalone, Protein Structure, Communities and Crime, Yeast, Mammographic Masses, Wine Quality, and Facebook Metrics. In Figure 6, the improvements of MIRACLE are minimal for MCAR (except for mean imputation). This agrees with our discussion in Section 2.1, because the baseline imputations are not biased in the MCAR setting where $\boldsymbol{X} \perp\!\!\!\perp_d \boldsymbol{R}$ holds in the $m$-graph. Conversely for the MAR and MNAR settings, as expected, we observe MIRACLE has an significant improvement on some of the datasets, such as Abalone, Autism, Energy and Protein Structure. As discussed in Section 2, MIRACLE can improve the baseline imputation under Assumptions 1-4, which may not hold in these real-world datasets. Nevertheless, we observe that MIRACLE never degrades performance relative to its baseline imputation on any dataset. Furthermore, no baseline imputer is optimal across the datasets. In almost all cases, applying MIRACLE to any baseline results in the lowest error. We show a similar gain for this experiment with MCAR and MNAR in Appendix C. An **ablation study** on our overall loss is provided in Appendix F.

---

[3]Source code at `https://github.com/vanderschaarlab/MIRACLE`.

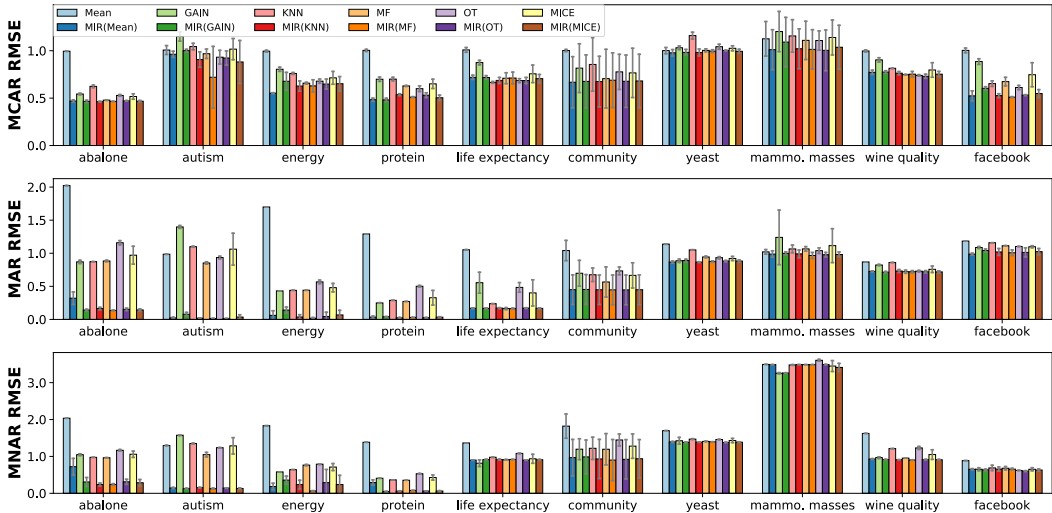

Figure 6: **MIRACLE on real data**. MIRACLE improves all baselines across MCAR **(top)**, MAR **(middle)**, and MNAR **(bottom)**. Results for predictive error and congeniality can be found in the Appendix C.

### 4.3    Causal discovery and imputation performance

In our experiments, we observe a positive correlation between the quality of learned DAGs (and causal parents) with imputation performance. Consider the left-most plot in Figure 7 using OT as a baseline imputer under MAR on our real data sets. Here, we do not have oracle knowledge but assume that the sparseness of the learned DAG implies a coherent DAG. We observe that MIRACLE has the most performance gain when fewer causal parents are identified for the missing variable in the learned DAGs. When MIRACLE is less able to isolate causal parents for prediction, the learned DAGs contains many spurious edges, and MIRACLE only has marginal improvements over the baseline imputer. We note that the gain of MIRACLE is not reproducible via feature selection methods, which are still prone to the spurious correlations in the observed data, as discussed in Section 2.1.

### 4.4    MIRACLE Convergence

In this subsection, we investigate two dimensions of MIRACLE refinement: (1) baseline imputation quality and (2) sample or instance-wise refinement. Regarding baseline imputation quality, we are interested in understanding the impact of MIRACLE refinement on various baseline imputers that may have disparate performances. In the middle plot of Figure 7, we show MIRACLE applied to various baseline imputers on the Abalone dataset. Similar plots for other datasets can be found in Appendix C. We observe that even though mean imputation starts off with the worst error, after refinement by MIRACLE, we see that all methods converge to similar RMSEs. For the second experiment, we investigate the sample-wise improvement of MIRACLE on the abalone dataset using MissForest as a baseline imputer. On the right-most plot of Figure 7, we observe that a vast majority of the samples are improved by MIRACLE. Note that every point below the diagonal is considered an improvement on an instance over the baseline imputation method. We can see MIRACLE improves the imputation almost universally except for the instances with small errors in the baseline imputation; on these instances, MIRACLE does not inflate their errors by a large margin. Furthermore, we observe that MIRACLE iteratively improves imputation as training progresses (over each epoch) by the observation that the slope of each line decreases with each epoch.

## 5    Discussion

In conclusion, motivated by the minimax optimization problem (1) arising from interventions on missingness indicators in the $m$-graph that encode the conditional independencies in the data distribution, we proposed MIRACLE, an iterative framework to refine any baseline missing data imputation to use the causal parents embodied in the estimated $m$-graphs. MIRACLE learns the causal $m$-graph as an adjacency matrix embedded in its input layers. We proposed a two-part regularizer based on the causal graph and a moment regularizer based on the missing variable indicators. We demonstrated

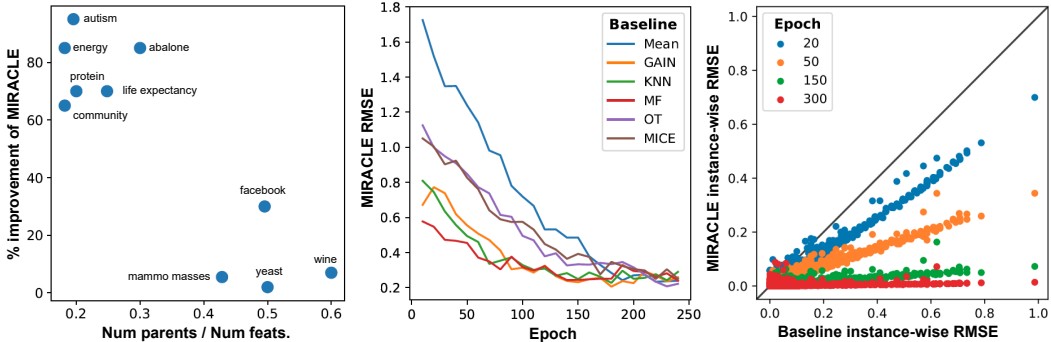

Figure 7: **(Left) Analysis of MIRACLE w.r.t. causal parents on real data.** MIRACLE has the most gain when we have identified a sparse set of causal parents. When many features are identified as causal parents, imputation performance degrades. **(Mid) Convergence of MIRACLE across various baseline imputers.** On the abalone dataset, we show that MIRACLE converges to consistent RMSE regardless of baseline imputation. **(Right) Sample-wise RMSE for MIRACLE across various epochs.** MIRACLE is applied to refine MissForest imputations, demonstrating that error is reduced in a sample-wise basis. Note: anything below the diagonal, is an improvement over the baseline imputations.

that MIRACLE significantly improved the imputations of six baseline imputation methods over a variety of synthetic and real datasets. MIRACLE never hurts performance in the worst-case, and we envision MIRACLE becoming a de facto standard in refining missing data imputation.

There are several limitations we would like to identify as paths for future work. First, any violation of the assumptions in Section 2 may adversely impact the performance of MIRACLE in practice. Second, causal discovery under missing data is an ongoing research area, and therefore MIRACLE may be discovering DAGs with bias introduced from the baseline methods. However, in experiments, MIRACLE still performs well even if it starts with mean imputation. We expect MIRACLE to improve as causal discovery methods under missingness improve. Third, in its current form, MIRACLE is not extensible to scenarios where causality may not be applicable, such as computer vision. Fourth, because of the causal discovery regularizer and network architecture, MIRACLE may have difficulty scaling to very high dimensional data. Lastly, a more general and detailed discussion is needed between our work and the merits of causality and robustness.

## Acknowledgements

This work was supported by *GlaxoSmithKline* (GSK), the *National Science Foundation* (NSF) under grant number 1722516, the *Office of Naval Research* (ONR), and *The Alan Turing Institute* (ATI). We thank all reviewers for their invaluable comments and suggestions.

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
