# Appendix

This appendix is outlined as follows:

- Section A details our synthetic data generating process and how we generated missingness.
- Section B contains additional experiments on synthetic data testing for imputation performance, testing for prediction performance of a downstream machine learning algorithm on imputed data, and testing for congeniality. In each case we consider MCAR, MAR and MNAR missingness patterns. We used the same experimental setup in [33] for testing prediction performance of a downstream machine learning algorithm and for congeniality.
- Section C contains additional experiments on real datasets testing for prediction performance of a downstream machine learning algorithm on imputed data, and testing for congeniality over all types of missingness.
- Section D provides an overview of our model and training details.
- Section E provides a computational complexity analysis of MIRACLE.
- Section F provides an ablation study for the MIRACLE loss function.
- Section G provides experiments regarding missingness location.

# A Supplementary experimental details

## A.1 Synthetic data generation

In each synthetic experiment, we generated a $p$-dimensional random graph $G$ from a Erdös–Rényi random graph model with $p$ edges on average. Given $G$, we assigned uniformly random edge weights to obtain a weighted adjacency matrix $W \in \mathbb{R}^{p \times p}$. Given $W$, we sampled $X = WX + E$ repeatedly from a Gaussian noise model for $E \in \mathbb{R}^p$ (each dimension sampled independently) to generate independent observations from this system.

## A.2 Generating missingness.

The following explains how we constructed synthetic datasets that satisfy MCAR, MAR and MNAR patterns of missingness. We apply a modification to the missingness generation from [33].

- **MCAR**. Missing completely at random was introduced by randomly removing $30\%$ of the observations in each feature.

- **MAR**. We sequentially define the probability that the $i$-th component of the $n$-th sample is observed conditional on the missingness and values (if observed) of the previous $i - 1$ components to be,

$$P^m(i) = \frac{p^m(i) \cdot N \cdot \exp(\sum_{j<i} w_j m_j(n) x_j(n) + b_j(1 - m_j(n)))}{\sum_{l=1}^{N} \exp(\sum_{j<i} w_j m_j(l) x_j(l) + b_j(1 - m_j(l)))} \tag{6}$$

  where $p^m(i)$ corresponds to the average missing rate of the $i$-th feature, and $w_j$, $b_j$ are sampled from $\mathcal{U}(0, 1)$ (but are only sampled once for the entire dataset).

- **MNAR.** Missing not at random was introduced by defining the probability of the $i$-th component of the $n$-th sample to be observed by,

$$P^m(i) = \frac{p^m(i) \cdot N \cdot \exp(-w_i x_i(n))}{\sum_{l=1}^{N} \exp(-w_i x_i(l))} \tag{7}$$

  with the same notation as above. Here, the missingness of a data point is directly dependent on its value (with dependence determined by the weight $w_i$, sampled from $\mathcal{U}(0, 1)$).

# B Additional synthetic results

In this section, we provide supplementary results for synthetic experiments. Note that the error bars are large for some of the plots with predictive error and congeniality. This is because the y-axis of these plots are min-max normalized between 0 and 1, so the high variance (large error bars) shows that the improvement by MIRACLE may be minimal for the mentioned datasets. Additionally, this could be caused by the fact that the missing features aren't predictive of a target variable, i.e., better imputation does not necessarily lead to any performance gain for the predicting the target variable.

## B.1 MCAR Results

Using our synthetic experimental setup used in the main paper, we show the performance of MIRACLE in terms of RMSE, predictive error, and congeniality in Figure 8 for each of our baseline methods with MCAR.

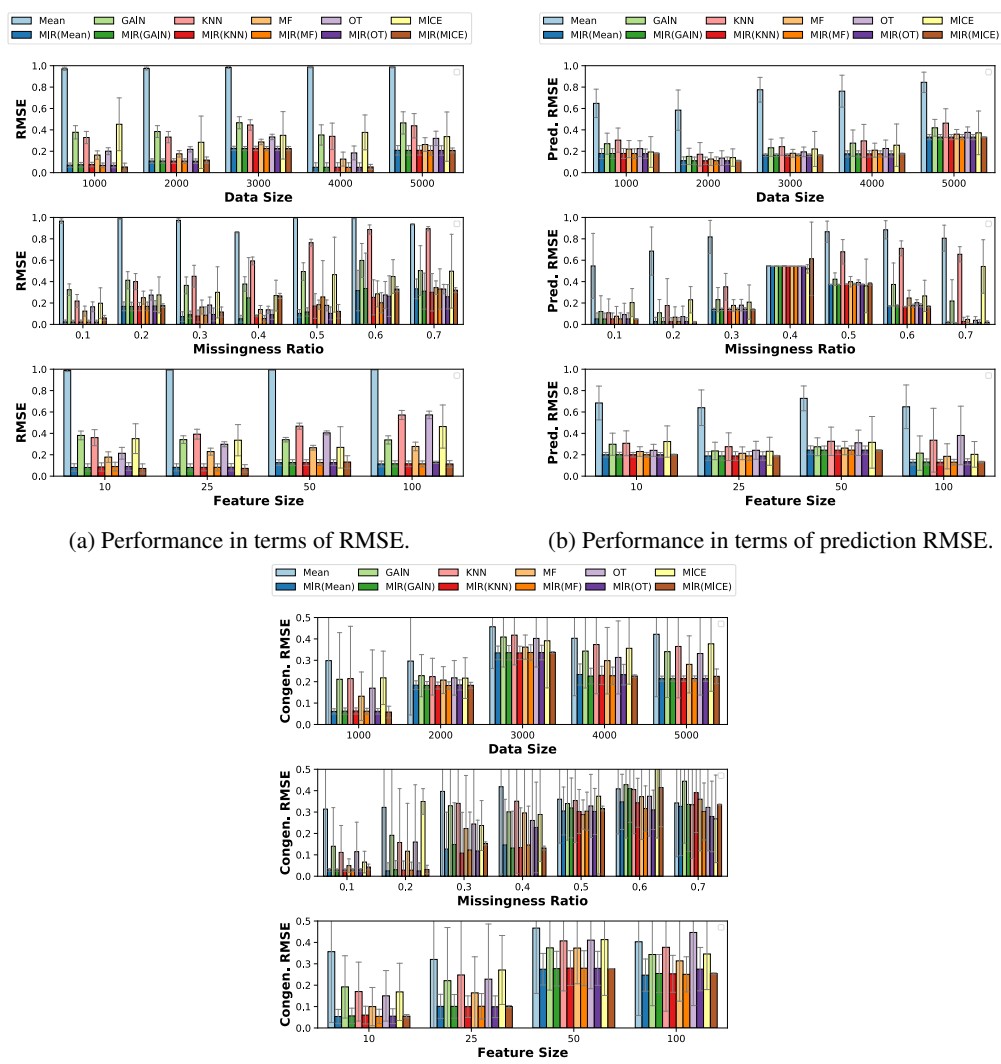

(a) Performance in terms of RMSE.

(b) Performance in terms of prediction RMSE.

(c) MCAR congeniality (in terms of RMSE).

Figure 8: Experiments on MCAR synthetic data as a function of dataset sizes **(top)**, missingness rates **(middle)**, and feature sizes **(bottom)** of each subfigure: **(a)** RMSE, **(b)** machine learning predictive error of a random variable, and **(c)** congeniality.

## B.2 MAR Results

Using our synthetic experimental setup used in the main paper, we show the performance of MIR-ACLE in terms of RMSE, predictive error, and congeniality in Figure 9 for each of our baseline methods with MAR.

## B.3 MNAR Results

Using our synthetic experimental setup used in the main paper, we show the performance of MIR-ACLE in terms of RMSE, predictive error, and congeniality in Figure 10 for each of our baseline methods with MNAR.

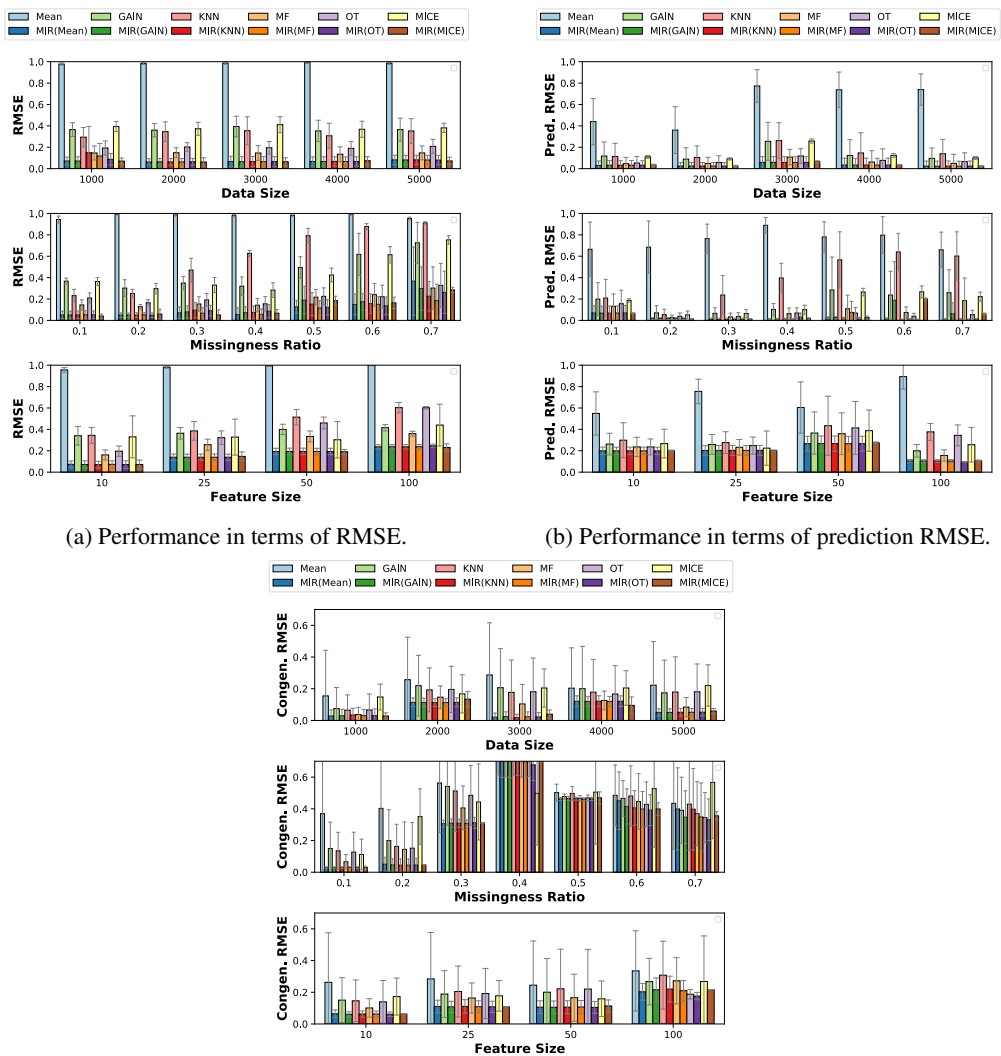

(a) Performance in terms of RMSE.  (b) Performance in terms of prediction RMSE.

(c) Congeniality (in terms of RMSE).

Figure 9: Experiments on MAR synthetic data as a function of dataset sizes **(top)**, missingness rates **(middle)**, and feature sizes **(bottom)** of each subfigure: **(a)** RMSE, **(b)** machine learning predictive error of a random variable, and **(c)** congeniality.

## C   Additional real datasets

**Prediction error and congeniality.**   We include additional plots for the real data experiments for prediction error and congeniality in Figures 12 and 13, respectively.

**Additional convergence plots.**   We include additional convergence plots on real datasets in Figure 11. We use the same experimental setup used in Figure 7 in Section 4. We observe that MIRACLE is able to converge regardless of baseline imputation used.

## D   Model and training details

We used the following network architecture for MIRACLE. Our proposed architecture consists of $d$ sub-networks with shared hidden layers, as shown in Figure 4. Each network is constructed with two hidden layers of $d$ neurons with ELU activation. Each benchmark method is initialized and seeded identically with the same random weights. For dataset preprocessing, all continuous variables are

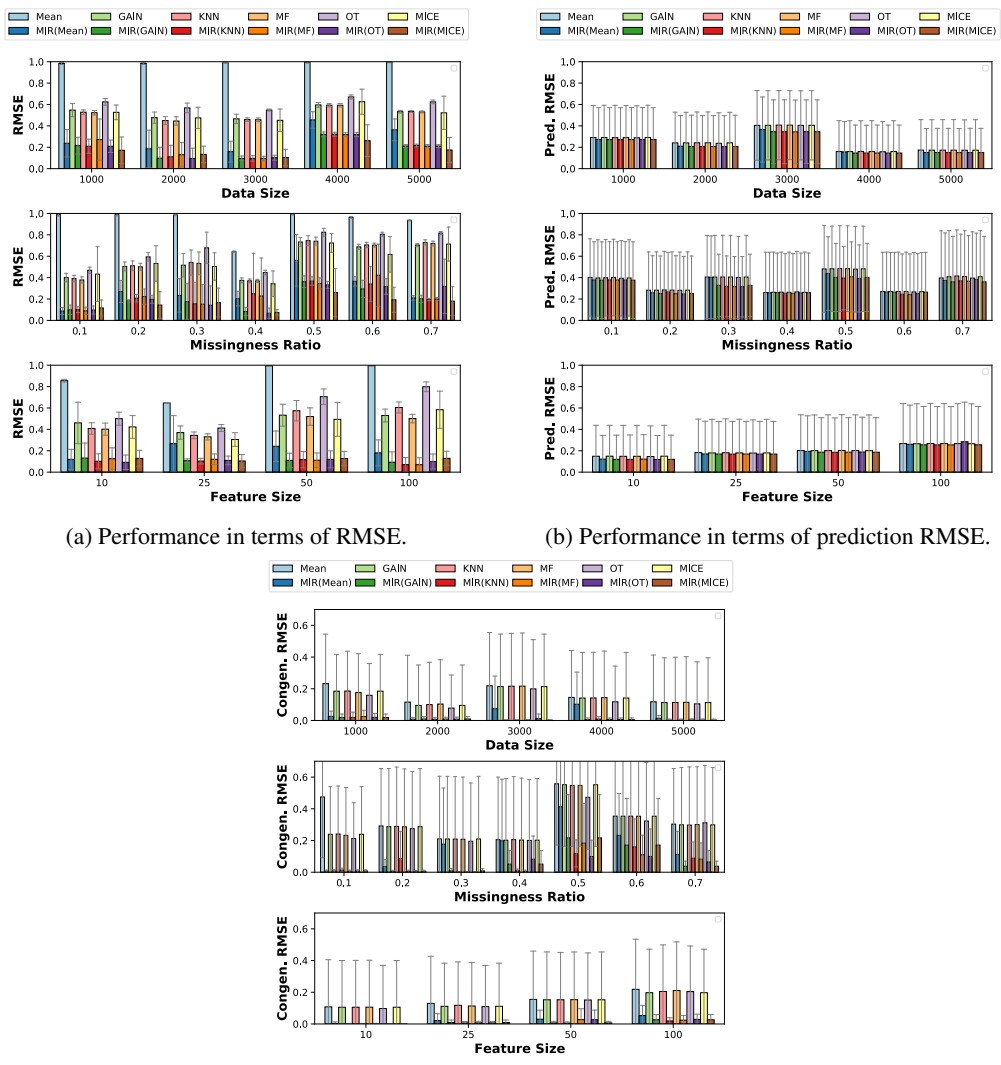

(a) Performance in terms of RMSE.

(b) Performance in terms of prediction RMSE.

(c) Congeniality (in terms of RMSE).

Figure 10: Experiments on MNAR synthetic data as a function of dataset sizes **(top)**, missingness rates **(middle)**, and feature sizes **(bottom)** of each subfigure: **(a)** RMSE, **(b)** machine learning predictive error of a random variable, and **(c)** congeniality.

standardized with a mean of 0 and a variance of 1. We train each model using the Adam optimizer with a learning rate of 0.0005 for up to a maximum of 300 epochs.

**Computational hardware.** All models were trained on an Ubuntu 18.04 OS with 64GB of RAM (Intel Core i7-6850K CPU @ 3.60GHz) and 2 NVidia 1080 Ti GPUs.

# E  Computational Complexity

Pseudocode for MIRACLE is provided in Algorithm 1. We perform an analysis of the MIRACLE scalability. Using our synthetic data generation, we created datasets of 1000 samples. Using our the synthetic experimental setup presented in the main paper, we present the computational timing results for MIRACLE as we increase the number of input features on inference and training time in Figure 14. Computational time scales linearly with increasing the number of input samples. As expected, we observe that the time to train 1000 samples grows exponentially with the feature size; however, the inference time remains linear. Inference time on 1000 samples with 400 features takes approximately 1.1 seconds, while training time takes nearly 85 seconds. Experiments were conducted on an Ubuntu 18.04 OS using 6 Intel i7-6850K CPUs.

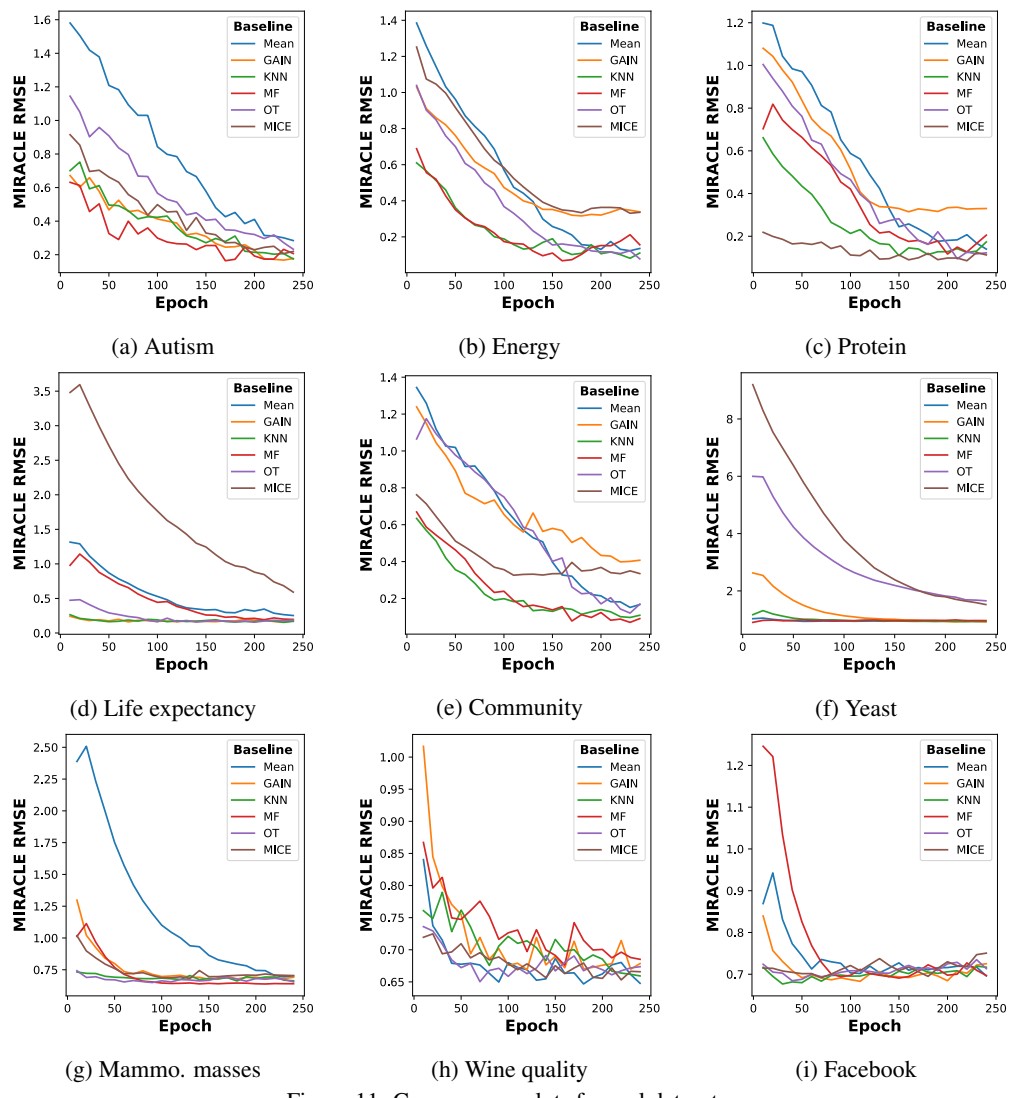

Figure 11: Convergence plots for real datasets.

---

**Algorithm 1** Train MIRACLE

---

**Input:** An incomplete dataset $\mathbf{X}$ with missing values, a missing indicator matrix $\mathbf{R}$ (with 1 indicating observed), an imputed matrix $\tilde{\mathbf{X}}^{(0)}$ by some baseline method,

**Output:** Imputed dataset $\tilde{\mathbf{X}}^*$ with no missing values.

**Initialization:** Imputation network $f$, $\mathcal{G} = \varnothing$ with maximum size $M_{\mathcal{G}}$ for saving imputed matrices over epochs

**repeat**

    Train $f$ for one epoch by optimizing the objective function $\mathcal{L} = \mathcal{L}_1 + \beta_1 \mathcal{R}_1 + \beta_2 \mathcal{R}_2$ with $\tilde{\mathbf{X}}^{(0)}$ as input.

    **if** $\mathcal{G}$ is full **then**

        Remove the first element from $\mathcal{G}$

    **end if**

    $\tilde{\mathbf{X}} \leftarrow f(\tilde{\mathbf{X}}^{(0)}), \mathcal{G} \leftarrow \mathcal{G} \cup \{\tilde{\mathbf{X}}\}$

    $\tilde{\mathbf{X}}^{(0)} \leftarrow$ average all the elements of $\mathcal{G}$.

**until** MIRACLE converges (i.e., change of $\mathbf{X}^{(0)}$ is small)

**return** $\tilde{\mathbf{X}}^* \leftarrow \tilde{\mathbf{X}}^{(0)}$

---

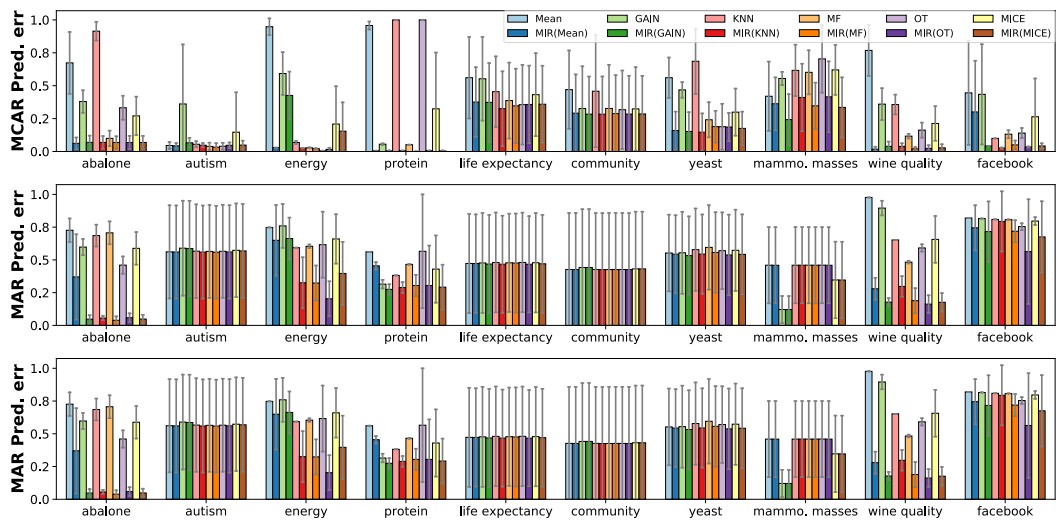

Figure 12: **MIRACLE on real datasets in terms of predictive error**. MIRACLE improves over all baselines across all types of missingness: MCAR **(top)**, MAR **(middle)**, and MNAR **(bottom)**.

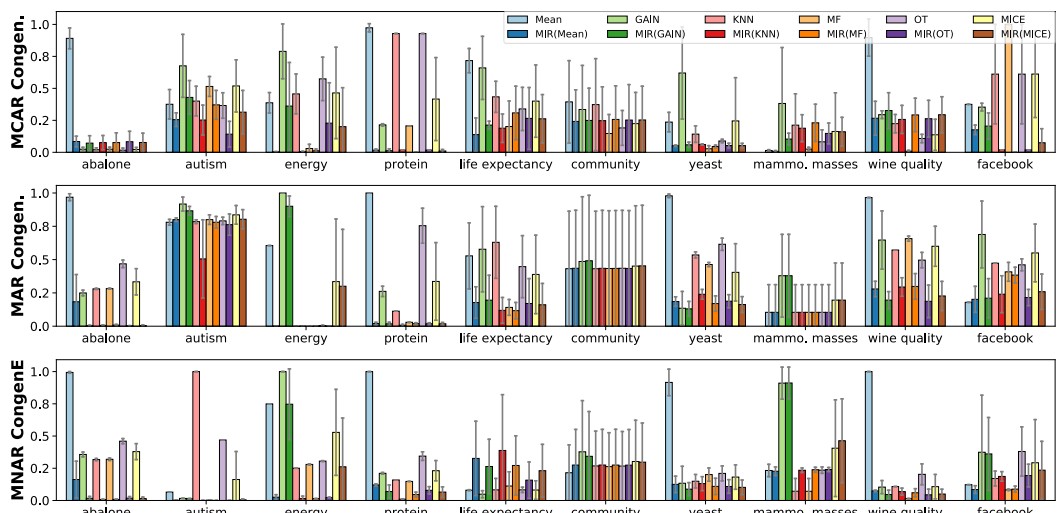

Figure 13: **MIRACLE on real datasets in terms of congeniality**. MIRACLE improves over all baselines across all types of missingness: MCAR **(top)**, MAR **(middle)**, and MNAR **(bottom)**.

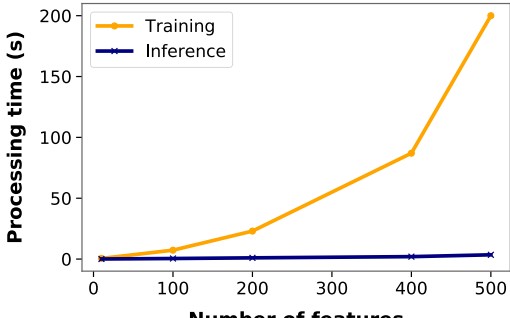

Figure 14: **MIRACLE scalability analysis**

## F  Ablation study

We provide an ablation study on our MIRACLE loss function in Eq. 5 to understand the sources of gain of MIRACLE. Here we execute this experiment on our real datasets using the same experimental details highlighted in the main manuscript. We show the results of our ablation on MIRACLE using MissForest as baseline imputation with MAR missingness to highlight our sources of gain in Table 1. Here, we observe that MIRACLE (rightmost column) has the most gain over all datasets. Additionally, we observe that $\mathcal{L}_1 + \mathcal{R}_1 + \mathcal{R}_2$ has the most gain when MIRACLE has the most performance improvement over the baseline (see Fig. 7 in the manuscript).

Table 1: **Ablation study of MIRACLE on real datasets to highlight sources of gain.**

| Dataset | $\mathcal{R}_1 + \mathcal{R}_2$ | $\mathcal{L}_1 + \mathcal{R}_2$ | $\mathcal{L}_1 + \mathcal{R}_1$ | $\mathcal{L}_1 + \mathcal{R}_1 + \mathcal{R}_2$ |
|---|---|---|---|---|
| abalone | $0.321 \pm 0.108$ | $0.521 \pm 0.199$ | $0.312 \pm 0.082$ | $0.222 \pm 0.062$ |
| autism | $0.093 \pm 0.005$ | $0.094 \pm 0.004$ | $0.091 \pm 0.004$ | $0.073 \pm 0.004$ |
| energy | $0.106 \pm 0.011$ | $0.147 \pm 0.077$ | $0.132 \pm 0.050$ | $0.065 \pm 0.061$ |
| protein | $0.134 \pm 0.016$ | $0.129 \pm 0.008$ | $0.119 \pm 0.010$ | $0.080 \pm 0.008$ |
| life expectancy | $0.239 \pm 0.007$ | $0.223 \pm 0.019$ | $0.216 \pm 0.014$ | $0.208 \pm 0.015$ |
| community | $0.490 \pm 0.015$ | $0.516 \pm 0.020$ | $0.479 \pm 0.023$ | $0.463 \pm 0.010$ |
| yeast | $0.984 \pm 0.013$ | $0.984 \pm 0.006$ | $0.988 \pm 0.004$ | $0.950 \pm 0.014$ |
| mammo masses | $1.105 \pm 0.010$ | $1.150 \pm 0.009$ | $1.103 \pm 0.013$ | $1.040 \pm 0.013$ |
| wine quality | $0.797 \pm 0.004$ | $0.745 \pm 0.013$ | $0.724 \pm 0.008$ | $0.722 \pm 0.003$ |
| facebook | $1.056 \pm 0.005$ | $1.032 \pm 0.044$ | $1.034 \pm 0.056$ | $0.983 \pm 0.002$ |

## G  Understanding missingness location

An important consideration is how well does predicting with the causal parents work when down-selecting features. Consider missingness in $X_5$ in the DAG in Figure 15. The first column with the causal parents $\text{Pa}(X_5)$ mean that only the parents of features were used for imputation. $X_9$ represents a variable that is not causally linked to anything.

Using our synthetic data generating process, we synthesized a dataset according to Figure 15. The goal here was to impute the missing values in $X_5$, using each variable in Figure 15 to induce the missingness. Each of the missingness causes is categorized as MAR, except for $X_5$, which is MNAR (since missingness caused by itself), and for $X_9$, which is MCAR, since it is an external noise variable. The results provide several interesting findings.

1. Using MissForest as a baseline imputer, the results in Table 2 show that MIRACLE performs as well as $\text{Pa}(X_5)$, and has better performance than the baseline imputer. Moreover, the two right-most columns of Table 2 give the average estimated functional dependence of $X_5$ (our target for prediction) and its parents and non-parents. We see that MIRACLE recovers true parents consistently.

2. We see that using causal parents ($\text{Pa}(X_5)$ and MIRACLE) for missingness caused by itself, $X_5\ddagger$, and a noise variable, $X_9\dagger$, leads to the least amount of improvement.

3. We see that MIRACLE has the most gain when the missingness is caused by a causal parent ($X_2$ or $X_3$).

4. Interestingly, for this example, we observe comparable performance when using the Markov blanket features versus all features in our baseline algorithm (MissForest). This suggests that the Markov blanket features are likely used for imputation by the baseline method.

In Appendix B, we show that the baseline performs similarly to the Markov blanket features colored in blue in Figure 15.

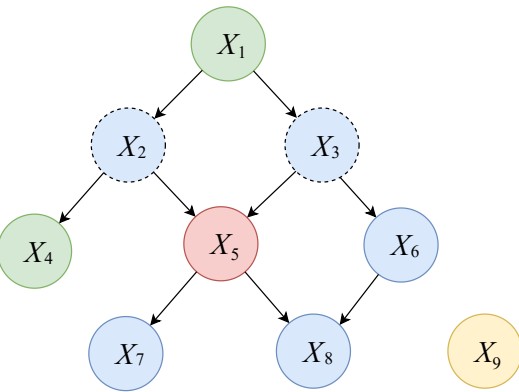

Figure 15: **A sample DAG.** $X_5$ is the incomplete variable in red. The Markov Blanket $\mathtt{MB}(X_5)$ is shown in blue, and the causal parents $\mathrm{Pa}(X_5)$ are shown with dashed borders. $X_9$ represents a variable that is not causally linked to anything.

Table 2: **Understanding the location of missingness**. We predict $X_5$ when its missingness is caused by each variable in the DAG. ‡ and † represent MNAR and MCAR, respectively. All other causes are MAR. The two right-most columns show the learned edge weights into $X_5$ for the parental and non-parental variables.

| | $X_5$ imputed error (RMSE) | | | | $X_5$ edge weights (no threshold) | |
|---|---|---|---|---|---|---|
| Cause | $\mathrm{Pa}(X_5)$ | $\mathtt{MB}(X_5)$ | Baseline | MIRACLE | Pa | non-Pa |
| $X_1$ | $0.11 \pm .06$ | $0.15 \pm .03$ | $0.27 \pm .05$ | $\mathbf{0.12 \pm .07}$ | $0.44 \pm 0.14$ | $0.02 \pm 0.01$ |
| $X_2$ | $0.98 \pm .08$ | $1.34 \pm .05$ | $1.31 \pm .06$ | $\mathbf{0.49 \pm .06}$ | $0.64 \pm 0.09$ | $0.01 \pm 0.01$ |
| $X_3$ | $1.20 \pm .04$ | $1.49 \pm .04$ | $1.45 \pm .09$ | $\mathbf{0.50 \pm .06}$ | $0.62 \pm 0.13$ | $0.02 \pm 0.01$ |
| $X_4$ | $\mathbf{0.69 \pm .05}$ | $1.20 \pm .07$ | $1.23 \pm .05$ | $1.04 \pm .05$ | $0.29 \pm 0.11$ | $0.13 \pm 0.05$ |
| $X_5$‡ | $\mathbf{1.51 \pm .03}$ | $1.75 \pm .08$ | $1.76 \pm .06$ | $1.59 \pm .07$ | $0.37 \pm 0.18$ | $0.03 \pm 0.02$ |
| $X_6$ | $\mathbf{0.13 \pm .08}$ | $0.17 \pm .04$ | $0.18 \pm .07$ | $0.14 \pm .05$ | $0.34 \pm 0.15$ | $0.05 \pm 0.02$ |
| $X_7$ | $1.04 \pm .05$ | $1.47 \pm .04$ | $1.47 \pm .06$ | $\mathbf{1.01 \pm .06}$ | $0.39 \pm 0.05$ | $0.04 \pm 0.01$ |
| $X_8$ | $0.21 \pm .04$ | $0.28 \pm .05$ | $0.23 \pm .03$ | $\mathbf{0.20 \pm .03}$ | $0.46 \pm 0.10$ | $0.02 \pm 0.01$ |
| $X_9$† | $0.15 \pm .03$ | $0.18 \pm .04$ | $0.17 \pm .07$ | $\mathbf{0.14 \pm .05}$ | $0.31 \pm 0.15$ | $0.02 \pm 0.01$ |

## H  Convergence

Using the same experimental setup, in Appendix G, we examine how the estimated adjacency matrix converges to the truth as sample size increases. Table 3, shows that as sample size increases we see that the quality of the learned m-graph improves.

Table 3: **Convergence of DAG weights as dataset size increases.** $\circ$ is the element-wise product and $\hat{W}$ is the predicted adjacency matrix weights

| Dataset size | $\sum_{i,j} (\hat{W} \circ W) / \sum_{i,j} \hat{W}$ |
|---|---|
| 100 | $0.135 \pm 0.04$ |
| 500 | $0.242 \pm 0.03$ |
| 1000 | $0.252 \pm 0.03$ |
| 5000 | $0.910 \pm 0.01$ |
| 10000 | $0.916 \pm 0.01$ |