# OpenReview forum: "MIRACLE: Causally-Aware Imputation via Learning Missing Data Mechanisms"
_NeurIPS.cc/2021/Conference — NeurIPS 2021 Poster_

### Official Review · Reviewer_4iEJ · 2021-07-05

**Rating:** 6
**Confidence:** 4

**Summary:**

The authors present MIRACLE, a neural network that learns to impute well thanks a specific regularization scheme. The proposed regularisation is motivated by the idea that when imputation is based on all observed data, spurious correlations can arise while it is not the case when it is based solely on the causal parents. Technically, the weights of the imputation network are regularised so that the dependencies between all imputation functions form a DAG. This is possible thanks to prior work which introduced a score based on adjacency matrices that is equal to zero if and only if the corresponding graph is a DAG.

The neural network proposed by the authors also includes a second regulariser (moment regularizer), this time aimed at matching two different estimators of the first moments of the data. Overall, it minimises a reconstruction loss combined with these two regularisers.

  Extensive experiments are conducted on simulated and real data that compare this neural network to various baselines in terms of imputation quality ( as well as performance on downstream tasks and congeniality in Supplementary materials). On simulated data, it is able to improve imputations over all baselines across all missing data mechanisms. On real data, it does not provide improvements in MCAR, but does so for half of the datasets in MAR and MNAR.


**Limitations And Societal Impact:**

-

**Main Review:**

**Strengths**:
- The idea of using a regularizer that promotes the dependencies among imputation functions to follow a DAG is original.
- Experiments are extensive. In the simulations, the number of samples, features, and missing data mechanisms (MCAR, MAR, MNAR) are varied. Experiments are also carried on 10 real datasets, with varied missing data mechanisms.
- The baselines include recent work (Gain and Optimal-transport based imputations) on top of classical baselines.
- The improvements in terms of imputation performance are clear in simulations and part of the real data.


**Weaknesses and questions**:
- Paragraph 3.3 (Causal Regularizer) is at the heart of the paper. Yet, it is very short and a few things are not clear to me.
This paragraph describes how the weights of the imputation network can be regularised so that the dependencies defined by f correspond to a DAG).
	- From my understanding, $\partial_k f_j$ is a number and not a vector. Is it the case? If yes the norm used around the partial derivative is a bit misleading.  Also the phrase ‘which can be obtained from the l2-norm of the k-th row…’ is unclear to me. It is a sufficient but non necessary condition for the derivative to be zero? And wouldn’t it be the column instead of the row?
    - Finally, the regularizer 1 (for ‘DAGness’) is about the adjacency matrix B but it is unclear to me what quantity is used for B_{k, j} as a function of the weights of the imputation network. I understand it is a quantity that depends on the weights of the first layer of $f_j$, but I do not understand exactly what quantity and why.
It would be helpful to explain this part a but more.

- It is unclear why MIRACLE works on MCAR simulated data while, as stated by the authors on the real data part, it is not expected to improve performance on MCAR data. On real data, MIRACLE does not improve results compared to baselines in MCAR. The authors explain that this finding agrees the discussion in section 2.1, which states that in MCAR the baselines are not biased. However, the simulations in MCAR in appendix show that MIRACLE brings improvement even if it is MCAR. It seems inconsistent at first sight with the explanation given for the real data. What could be the explanation for this?

- Some important details about the experiments are not given.
    - A downside of using two regularisers is that two hyperparameters need to be set. How are they set in the experiments?
    - What sample sizes are used for the train and test sets?
    - In the MAR and MNAR simulations, I understand that 30% of features have 30% missingness, is it correct? That would lead to a much smaller amount of missing values than in MCAR, in which I understand all features have 30% missing values.

-  I think it is important to understand the relative importance of the two regularizers, which are very different in nature. It is nice to have the ablation study for the real data, however I think it would give more insights into the relative importance of the regularisers on the simulated data. Notably, should we expect that the moment regularizer, which is derived for the MAR case and not appropriate for the MNAR case, help in the MNAR case?


**Time Spent Reviewing:**

5

---

> ### Author Response · Authors · 2021-08-09
> **Response to Reviewer 4iEJ**
>
> Thank you for your thoughtful comments and suggestions.
>
> **[Paragraph or Subsection 3.3.]**  We are very sorry for the typos and confusion.
>
> $\partial_k f_j$ is indeed a number. By “What can be obtained from $l_2$-norm of the k-th...”, we mean if the $k$-th column of the input layer weight matrix $\mathbf{ W_{1,j}^{imp}}$ or $\mathbf{W_{1,j}^{miss}}$ is 0, the network $f_j$ will not depend on the $k$-th variable, i.e.,  $\partial_k f_j = 0$. In a overparameterized fully connected neural network model $f_j$, "the $k$-th column of  $\mathbf{W_{1,j}^{imp}}$ or $\mathbf{W_{1,j}^{miss}} = 0$"   is not strictly necessary but the only realistic solution for achieving $\partial_k f_j = 0$ without over-reducing the approximation power of the network (e.g., setting the entire output layer to zero).
>
> In the adjacent matrix $\mathbf{B}$, $\mathbf{B_{k,j}}$ is computed as the $l_2$ norm of the $k$-th column of the input matrix $\mathbf{W_{1,j}^{imp}}$ or $\mathbf{W_{1,j}^{miss}}$. If $\mathbf{B_{k,j}}$ is zero, the generating function of the $j$-th variable, $ f_j$, will not depend on the $k$-th variable. In other words, the $k$-th variable is not a parent of the $j$-th variable in a directed acyclic graph (DAG). The causal regularizer $\mathcal{R_1}$ in our equation (3) ensures that the dependence structure of the adjacent matrix $\mathbf{B}$ follows a directed acyclic graph. We will provide a more preliminary and detailed introduction of the causal regularizer $\mathcal{R_1}$ in the revised manuscript. Thank you for your suggestion.
>
> **[MCAR Simulations]** In our Figure 6, the improvements of MIRACLE are minimal for MCAR (except for mean imputation) over ten different real-world datasets. As a direct comparison on the same datasets in the same figure, MIRACLE can improve the baselines significantly for MAR and MNAR where spurious correlations occur due to the missingness mechanisms as discussed in Section 2.1.
>
> In Appendix B, MIRACLE improves for MCAR, MAR and MNAR on the synthetic data generated via a linear gaussian structural equation model i.e. a linear DAG (see Appendix A). Given such a clear DAG structure, our causal regularizer $\mathcal{R_1}$ in equation (3), if used, is expected to generally improve the performance on top of any baselines which do not leverage the DAG structure. This gain is orthogonal to the gain from being robust towards spurious correlation.
>
> Thank you for your insightful comment. In the setting of MCAR with no spurious correlation, by comparing the performance of MIRACLE on synthetic and real-world datasets, we will emphasize a simple fact in the revised manuscript: For MCAR, MAR and MNAR, with the causal regularizer, our method MIRACLE can generally improve the performance if the data is indeed generated via a DAG. However, there is no guarantee the DAG assumption holds strictly on real-world datasets. In this case, our method has the potential to improve imputation performance by being robust towards suspicious correlation in the MAR and MNAR settings, but not the MCAR setting.
>
> **[Experimental details]** We apologize for not including some of the following details and will clarify them in the revised manuscript.
>
> 1. We performed a hyperparameter sweep (log-based) for $\beta_1$ and  $\beta_2$ with ranges between 1e-3 and 100.  By default we have  $\beta_1$ and  $\beta_2$ set to 0.1 and 1, respectively.
>
> 2. We used an 80/20 train/test split.
>
> 3. You are correct that in the MCAR setting there are much more missing features, but the important thing to consider here is that the missingness per feature rate is preserved (30%).  We believed this was the most important thing to sweep over since we were looking to learn the missing data mechanism at a feature-wise level.
>
> **[Two regularizers]** We have provided an ablation study of the two regularizers in Appendix F on ten different real-world datasets in the MAR setting. In this study, we have shown that adding the causal regularizer $\mathcal{R_1}$ can improve the performance over not using any regularizers; Further, adding the moment regularizer $\mathcal{R_2}$ on top of $\mathcal{R_1}$ always gives the best performance over the study. This result agrees with your comment that “the two regularizers are different in nature” as well as demonstrate that their performance gains are orthogonal in the MAR setting. As mentioned in Section 3.4 (lines 187- 189), the moment regularizer $\mathcal{R_2}$ is not suitable for MNAR. We should not expect $\mathcal{R_2}$ to improve the performance in the MNAR settings. In the revised manuscript, we will provide more simulations to further verify the relative importance of the regularizers in different settings (MCAR and MNAR). Thank you for your suggestions.

---

> > ### Comment · Reviewer_4iEJ · 2021-08-30
> > **Thank you for your clear answers**
> >
> > Thank you for your clear answers.
> >
> > While reading your manuscript I did not understand how the causal regularizer was defined, now I do. I think the further explanations you provided would really be useful in a revised version.
> >
> > I also understand your point about the MCAR simulations. I think the answer is interesting and contributes to a better understanding of why and when MIRACLE works.
> >
> > For these reasons, I upgraded my score.
> >
> > Otherwise, in terms of limitations:
> >  * As I mentioned in my review I think the ablation study on the simulated data would be helpful, in particular to verify that $\mathcal R_2$ has no positive effect in MNAR, but that it does otherwise.
> > * The authors show that MIRACLE converges on the proposed experiments. However, it seems to me that there is no argument to say that it always converges.
> > * Minor comment: I agree with the other reviewers that a quantification of how well the m-graphs are recovered on the simulated data is a valuable addition.

---

> > > ### Author Response · Authors · 2021-08-30
> > > **Thank you!**
> > >
> > > Thank you for your time and consideration! We really appreciate you getting back to us with an improved score.  On our end, we will address the limitations you pointed out in the revision as follows:
> > > - We will do an ablation study to verify that $\mathcal R_2$ has no positive effect in MNAR, but that it does otherwise.
> > > - We provided some justification regarding convergence in lines 210-218 with some theoretical references [4, 22, 31]. In the revision, we believe we can and will expand this discussion in the Appendix to theoretically prove the convergence of our proposed method using the ideas and techniques adopted from [4, 22, 31].
> > > - In our response to reviewers (qmy6 and c6qR), we have provided preliminary experimental results on the simulated data regarding how well the m-graphs are recovered. We will provide more comprehensive experiments in the revision.
> > >
> > > We sincerely thank you for your time and comments which we believe have significantly helped to improve our paper!

---

> ### Author Response · Authors · 2021-08-25
> **Additional comments**
>
> Thanks again for reviewing our paper.  Please let us know if there is anything else we can provide to help improve our score.  We believe that we have addressed the concerns mentioned in your review, and are happy to clarify further should you remain unsatisfied.
>
> Best, Authors of #4694

---

### Official Review · Reviewer_XE6q · 2021-07-12

**Rating:** 9
**Confidence:** 3

**Summary:**

This paper presents a causally-aware imputation algorithm (MIRACLE) which refines imputation of a baseline by simultaneously modeling the missingness generating mechanism. Extensive experiments on synthetic and a variety of publicly available datasets with several baseline comparisons are presented.

**Limitations And Societal Impact:**

Limitations not included.
Authors are suggested to identify limitations of causal graphs and the types of data that can be viewed in such modeling approaches.

**Main Review:**

*Very well done paper that leverages the modeling of the missing data mechanism. Along with knowledge of a causal graph, this kind of understanding of the data generating process is strong and should be condoned in data analytic systems.
*The paper is very thorough, figures are clear and elucidate well
*Very through experiments are presented, systematically varying feature and data size
*It would be helpful to see more discussion on the premise, that causal solutions are expected to correct for spurious correlations introduced by distribution shift due to missing data and preserve the dependencies of the complete data for downstream analysis.  If spurious variables are included (e.g. as parent variables), which can happen in data analyses, then how would those be de-coupled?

**Time Spent Reviewing:**

2

---

> ### Author Response · Authors · 2021-08-09
> **Response to Reviewer XE6q**
>
> We thank you for your thoughtful comments and suggestions.
>
> **[Spurious variables]** As a general observation, note that the DAG constraint, by enforcing a well-defined causal diagram over the system of variables, decouples the contribution of causal and confounded or spurious correlations in data forcing the imputation to rely on causal parents only. In a nutshell, for each one of the missing data points, in variable $X_i$, its parents $\text{Pa}(X_i)$ and  $X_{−i}$ (all the variables except  $X_{i}$), we aim at recovering $E[X_i \mid \text{Pa}(X_i)]$ instead of learning the full conditional $E[X_i \mid X_{-i}]$, prone to bias because of open paths of correlations through missingness indicators. Our example in Figure 1, describes this intuitively, and more details are included in Section 2.1 explaining why we may expect $E[X_i\mid  \text{Pa}(X_i)]$ and $E[X_i \mid X_{−i}]$ to differ. In the revised manuscript, we will expand our discussion on spurious correlations as you requested.
>
> **[Limitations]** Thank you, note that some discussion on the topic of limitations is included in Section 5, but more can be said. Imputation methods are to be understood in depth because imputation errors propagate to downstream tasks with often unknown consequences on the final data analysis task. MIRACLE is no different, especially since it relies on several assumptions for the correctness of inference. For instance, in general, we cannot expect to recover causal interactions and the target imputation function if some unobserved variables influence the system. From a societal perspective, MIRACLE is a tool designed to recover missing entries in data, and therefore we do not foresee any obvious, direct consequences for human well-being nor any large detrimental effects on its use in any safety-critical machine learning applications.

---

### Official Review · Reviewer_c6qR · 2021-07-16

**Rating:** 7
**Confidence:** 2

**Summary:**

This paper presents a new framework to impute missing values while preserving the causal structure of the data. The algorithm works by refining the imputations of a baseline method and simultaneously learning the missingness generating mechanisms (which cover different scenarios such as MCAR, MAR, MNAR). The proposed method obtains improved results when compared to the baselines in both synthetic and real experiments.

**Limitations And Societal Impact:**

Yes

**Main Review:**

In general, I think this is a good paper. It addresses a very timely problem of great interest to the machine learning community, i.e. the interplay between missing data imputation and the causal structure of the dataset (including the missingness indicators). The paper is well written and motivated. The proposed methodology is clear (in particular Figure 4 is useful to understand the network and optimization process), it seems sound, and it is different to previous approaches. Also, the experimental validation is well structured and is convincing.

Perhaps what I most miss is a deeper evaluation of the estimated m-graph. The focus of the experimental section is to show that MIRACLE reduces the test RMSE compared to the baseline. Only Figure 7 (left) presents a small discussion on the causal relationships found, although it is still linked to the improvement in terms of RMSE. My suggestion would be to analyse the m-graph estimated in the synthetic experiment, and compare it with the available ground truth. Many interesting questions can be answered there (e.g. are the causal relationships well reconstructed? is the causal regularizer really having an effect on the graph?).

Some other questions and comments that seem relevant:

* MIRACLE is a framework to refine an _existing_ imputation method by learning the causal structure of the dataset. I wonder whether it would be possible to develop (or whether it exists) some algorithm that provides the same functionality but _directly_, i.e. not starting by an existing imputation method that may introduce some bias. That is, I am thinking of an algorithm that simultaneously learns two things: 1) the causal structure of a dataset and 2) how to impute missing values.

* I was wondering why $[B]_{k,j}$ is defined like that in lines 173-174. I understand that, by doing the partial derivative, you are checking whether the j-th component depends on the k-th variable. And this is indeed what you want for the corresponding item of the adjacency matrix. If that is the case, it may be worth stating it explicitly.

* In the experimental section, there are some incorrect references to the Appendix. For instance, in the last sentence of Figure 5, it says "See Appendix B for MCAR and MNAR results". However, MNAR results are indeed in Figure 5. This wrong references happen several times throughout the experimental section, so please check it.

UPDATE

After reading the rest of reviews and the authors response, I think that this paper is already in good shape to be accepted. Therefore I keep my score (7).

**Time Spent Reviewing:**

5

---

> ### Author Response · Authors · 2021-08-09
> **Response to Reviewer c6qR**
>
> Thank you for your thoughtful comments and suggestions.
>
> **[m-graph estimation]** Thank you for your suggestions to analyze the estimated m-graph. This is an important analysis that we considered as well. The analysis in Figure 7 is not based on m-graphs because the analysis is conducted on the real-world datasets for which we did/do not know the true m-graphs. Nevertheless, as you said, Figure 7 shows a correlation between finding a causal relationship and improving imputation performance for our method. We agree that it is important to analyze the m-graph estimated in the synthetic experiment. Intuitively, if we learn the m-graph more accurately, we will impute the feature more accurately, because the missing features are part of the m-graph. In what follows, with experiments on synthetic data, we first answer your question by showing that the causal regularizer can reconstruct the m-graph as sample size increases, then we analyze if our causal imputation strategy works given any location (i.e. variable) in the m-graph that causes the missingness.
>
> In the table below, using synthetic data generated by the m-graph with nine variables ($X_1,\dotsc, X_9$) (see Figure 15 in Appendix G), we show the quality of the learned m-graph improves as we increase the dataset size.  Note that the values are not perfectly 1, as there will always be experimental variance in the data generation process. In Appendix G, we have analyzed different locations of missingness on the same synthetic dataset. We fix $X_5$ to the missing variable and let one of the other variables cause the missingness of $X_5$. We show that no matter which variable causes the missingness of $X_5$, imputing $X_5$ using its Markov blanket of $X_5$ is suboptimal, and imputing $X_5$ by our method or the true causal parents of $X_5$ is always optimal.  Lastly, the impact of the causal regularization in improving the imputation performance on real-world datasets was also shown as an ablation study in Table 1 of Appendix F (see the improved performance of $\mathcal{L_1} + \mathcal{R_1}$ and $\mathcal{L_1} + \mathcal{R_1} +  \mathcal{R_2} $).
>
> ===================================
>
> Dataset Size    |    sum(W * W_true) / sum(W)
>
> ===================================
>
> $\ \ \ \ \ \    $ 100      $\ \ \ \ \ \   $         |     $\ \ \ \ \ \ \ \ \ \    $     0.135 (0.04)
>
> $\ \ \ \ \ \    $ 500       $\ \ \ \ \  \   $        |     $\ \ \ \ \ \ \ \ \ \      $      0.242 (0.03)
>
> $\ \ \ \ \     $ 1000      $\ \ \ \ \     $       |     $\ \ \ \ \ \ \ \ \ \     $      0.252 (0.03)
>
> $\ \ \ \ \     $ 5000     $\ \ \ \ \     $        |      $\ \ \ \ \ \ \ \ \ \      $     0.910 (0.01)
>
> $\ \ \ \     $ 10000     $\ \ \ \     $      |     $\ \ \ \ \ \ \ \ \ \      $      0.916 (0.01)
>
> ===================================
>
>
> **[End-to-end framework]** Thank you for your interesting suggestion on building an end-to-end framework for causal imputation. In general, model-based imputation algorithms need to start with some initial imputation, such as random imputation, mean imputation or binary masking, because we can never feed an incomplete data matrix into a model. As shown in our experiments (e.g. the blue columns in Figures 5 and 6), our method can start with mean imputation and improves it significantly to be almost as good as the best benchmark in all cases. We argue that mean imputation is neutral and relatively less biased. However, if there is a baseline with high imputation accuracy, then its imputation can provide our method with a better initialization. An analogy of this can be made in neural network training. Normally, we train a neural network by initializing the network weights with mean 0. But sometimes we are given some pre-trained network, the question is if we should start with that pre-trained network instead of random initialization. We believe the answer to these questions are dependent on the data and prior knowledge if the baseline is high-quality or not. With MIRACLE we propose to iteratively refine the imputation using a *bootstrap strategy* in Section 3.5. What you propose (an algorithm that simultaneously learns two things: 1) the causal structure of a dataset and 2) how to impute missing values) is an interesting one. We believe this is a promising direction to explore that has the potential to promote faster convergence of causal imputation.
>
> **[$[ \mathbf{B} ]_{k,j}$ in lines 173-174]** By the definition of $[\mathbf{B}]_{k,j}$ in lines 173-174, we check whether the function $f_j$ (that generates the $j$-th variable) depends on the $k$-th variable or not. This is indeed what we want for the corresponding item of the adjacency matrix. We will make this clear in the revised manuscript.
>
> **[Typos]** Thank you very much for pointing out the incorrect references in our experimental section. We will double-check and correct all of them in the revised manuscript.

---

### Official Review · Reviewer_qmy6 · 2021-07-17

**Rating:** 5
**Confidence:** 4

**Summary:**

This paper proposes a new method for refining existing baseline missing data imputation algorithms. The method works by starting from an initial estimate from another baseline algorithm, then pass this estimate to two neural networks for refined estimation. The two neural networks are regularized based on intuitions from causal inference, such that the causal missingness of the data can be captured. The refinement process is iteratively run until convergence. The contribution of the paper is a new method for missing data imputation inspired by causal inference concepts.

**Limitations And Societal Impact:**

- On the computational complexity, I think the paper doesn't address it well. There are some discussions in Appendix E., "Computational time scales linearly with increasing the number of input samples.". However, a more important aspect, from my point of view, is the complexity w.r.t to the feature size. Since a matrix exponential is used in 3.3 Causal regularizer, the complexity is $O(d^3)$, and it would probably be very challenging to apply in a large scale applications.

**Main Review:**

+ The proposed method is inspired by causal inference, and this is less explored in the literature. Successful application of causal inference to missing data would be a significant step.

+ The paper ran extensive experiments and overall the benefits of the algorithm is demonstrated (see some minor points below)

+ The paper is well organized and easy to follow.

- I am a little bit concerned about the identifiability problem. As the paper mentioned in Line 205 itself, "Discovering a causal graph requires complete data. However, this is not the case for missing data." From my understanding, identifying causal structures requires access to interventions data, and I am not sure how the proposed approach connects to the identifiability problem, and also, how the four assumptions help define the identifiability. For example, is any theorems from [18] that could help explain why the proposed method can help recover the missing data/ missing mechanism? I think the paper lacks detailed discussion on this part.

- Following the previous point, if we are trying to recover the causal structures, is it possible to have some simulation studies to understand how well we are approximating the underlying causal graph? I think Figure 2 is trying to show the adjacency graph is improved over the course of the training. When we are synthesizing the missingness of the data, it appears possible to report how well we are learning the underlying graph by using the matrix B and compare to our synthesis.

- (Minor) It would be helpful for readers if the three missing data mechanisms are explicitly discussed.

- (Minor) The paper discussed about robust optimization in the introduction, but I think this discussion is forgotten in the later sections. How does the proposed approach connects to robust optimization is an interesting topic.

- For the experiments part, some of the results have really large error bars, for example, Figure 10. b), Figure 12 [life expectancy, community, yeast] in Appendix, and looks like the results would not be significant if you compare the baseline vs baseline + MIRACLE. I am wondering whether this provides any insight on the proposed algorithm

[18] Karthika Mohan, Judea Pearl, and Jin Tian. Graphical models for inference with missing data. In C. J. C. Burges, L. Bottou, M. Welling, Z. Ghahramani, and K. Q. Weinberger, editors, Advances in Neural Information Processing Systems, volume 26, pages 1277–1285. Curran Associates, Inc., 2013.

**Time Spent Reviewing:**

2

---

> ### Author Response · Authors · 2021-08-09
> **Response to Reviewer qmy6**
>
> We thank you for your thoughtful comments and suggestions.  We address each of your comments in turn.
>
> **[Identifiability]** By way of preface, note that the proposed approach infers the causal diagram that corresponds to the model that minimizes mean squared errors subject to a constraint on acyclicity. Concerning identifiability we make two remarks: (1) many data generating mechanisms are in fact identifiable, and (2) enforcing acyclicity serves as a useful constraint for imputation in general.
>
> Let $\text{Pa}(X_i)$ denote the parents of the $i$-th variable $X_i$ and let $E_i$ denote the error variable from random noise. When a model for the data takes the general form $X_i = f_i(\text{Pa}(X_i)) + E_i, i = 1, . . . , d$, the graph $G$ is not always uniquely defined: one example being $X = (X_1,\dotsc,X_d)$ jointly normally distributed and $f_i$ linear functions, in which case MIRACLE recovers a member of an equivalence class of DAGs that encodes the same conditional independencies as the true model (i.e. the data distribution). However, this case is somewhat exceptional and $G$ in many other models are provably uniquely identifiable with MIRACLE. For instance, if the $f_i$ are linear with non-Gaussian errors or with known error variances (see e.g. [A]), or if the functions f are nonlinear (see e.g. [B] and [C]), the graph $G$ is identifiable.
>
> These results still hold for the m-graphs with data missing completely at random (MCAR) although missing data may introduce spurious correlations otherwise (see [29] in the paper for related results). Under our assumptions, we can identify and estimate the missing data mechanism accurately in the MAR setting. The identifiability result above extends to the MAR setting. MAR is as easy as MCAR if we can estimate the missing data mechanism accurately. In the MNAR setting, the missing data mechanism is not uniquely identifiable, then the m-graph is not uniquely identifiable. In the revised manuscript, we will provide a detailed discussion regarding the identifiability issue in different settings. In experiments, we found that by recovering a member of an equivalence class of m-graphs that encodes the same conditional independencies (CIs) as the true model (i.e. the data distribution), i.e., overcoming spurious correlation due to non-existing CIs, our causal (acyclicity) regularizer $\mathcal{R_1}$ in equation (3) is a useful constraint for imputation of missing data that improves over the benchmarks that do not leverage causality in both the MAR and MNAR settings (See Figures 5 and 6 in the main text).
>
> [A] P.-L. Loh and P. B ̈uhlmann. High-dimensional learning of linear causal networks via inverse covariance estimation. Journal of Machine Learning Research, 15:3065–3105, 2014.
>
> [B] K. Zhang and A. Hyv ̈arinen. On the Identifiability of the Post-Nonlinear Causal Model. In Uncertainty in Artificial Intelligence, 2009.
>
> [C] P. O. Hoyer, D. Janzing, J. M. Mooij, J. Peters, and B. Sch ̈olkopf. Nonlinear causal discovery with additive noise models. In Advances in neural information processing systems, pages 689–696, 2009.
>
> **[m-graph discovery]** Yes, this is definitely possible and as the reviewer rightly points out Figure 2 obtained from one of our experiments hints at this property of MIRACLE. Furthermore, To provide more evidence on this property, In the table below, using synthetic data generated by the m-graph with nine variables ($X_1,\dotsc, X_9$) (see Figure 15 in Appendix G), we show the quality of the learned m-graph improves as we increase the dataset size.  Note that the values are not perfectly 1, as there will always be experimental variance in the data generation process. In the revised manuscript, we will add a visualization of how the estimated adjacency matrix B converges to the truth as the sample size increases.
>
> ===================================
>
> Dataset Size    |    sum(W * W_true) / sum(W)
>
> ===================================
>
> $\ \ \ \ \ \    $ 100      $\ \ \ \ \ \   $         |     $\ \ \ \ \ \ \ \ \ \    $     0.135 (0.04)
>
> $\ \ \ \ \ \    $ 500       $\ \ \ \ \  \   $        |     $\ \ \ \ \ \ \ \ \ \      $      0.242 (0.03)
>
> $\ \ \ \ \     $ 1000      $\ \ \ \ \     $       |     $\ \ \ \ \ \ \ \ \ \     $      0.252 (0.03)
>
> $\ \ \ \ \     $ 5000     $\ \ \ \ \     $        |      $\ \ \ \ \ \ \ \ \ \      $     0.910 (0.01)
>
> $\ \ \ \     $ 10000     $\ \ \ \     $      |     $\ \ \ \ \ \ \ \ \ \      $      0.916 (0.01)
>
> ===================================
>
> **[Explain the three missing data mechanisms]** In the current manuscript, we discussed the missing data mechanisms using conditional independencies in the m-graphs. We will add a more preliminary and explicit discussion about the mechanisms in the revised manuscript. Thank you for your great suggestion.
>
> **[Robust optimization]** Robust optimization is a powerful motivation for learning causal relationships as these tend to be invariant to domain changes driven by interventions in the underlying data generating mechanism. This connection is well established, and our observation in the context of imputation is to note that the shift between incomplete and complete data may also be interpreted as a change driven by interventions on missingness indicators. Our objective in this paper however is on an algorithmic solution to the imputation of missing data as well as introducing causality as regularization for imputation problems. A more general and detailed discussion on the connection between causality and robustness merits articles on its own, and lies outside of the purview of this paper where we focus. Thank you for suggesting the discussion, we will mention this as a future direction in our revised paper.
>
>
> **[Experiments]** The error bars are indeed larger for these plots. However, we would like to point out that Figure 10 (b) and 12 are for the alternative metrics of congeniality and predictive error, respectively. By predictive error, we are referring to the downstream prediction task using the imputed data.  Note that the y-axis of these plots are min-max normalized between 0 and 1, so the high variance (large error bars) shows that the improvement by MIRACLE may be minimal for the mentioned datasets.  Your comment points to an interesting and common problem in assessing data imputation methods - that is - if the missing features aren’t predictive of a target variable, better imputation for them does not lead to any performance gain in predicting the target variable i.e., lower predictive error. For example, Figure 10 (a) shows our imputation method has lower RMSE in imputing the missing features while Figure 10 (b) shows that our method does not lead to any gain in predicting the target variable.  We believe that the imputation RMSE is still the most transparent and rigorous metric for assessing imputation methods.  In the revised paper, we will discuss the problem of predictive error in method evaluation, and be sure to clarify that some of the large error bars are products of a combination of normalization mechanics and evaluation strategy.
>
> **[Scalability]** We appreciate the comment regarding scalability.  We agree that if the dataset is very high-dimensional data (say $d>5000$), MIRACLE will be computationally expensive due to the calculation of the matrix exponential. However, consider the fact that inverting the covariance matrix to compute the least square solution of a linear regression model is also $O(d^3)$, which often finishes in seconds as long as the dimension $d <= 5000$ (even 10000) on a normal CPU. MIRACLE is very scalable for datasets with moderate dimensions ($d<1000)$. In Appendix E, we have provided an experimental analysis of training complexity in terms of the dimension $d$. We can see the training time of our model with $d= 500$ and sample size $= 1000$ is around $200$ seconds using 6 Intel i7-6850K CPUs. From a practical standpoint, data imputation is typically considered an offline prediction task, where the primary objective is imputation accuracy. In the revised manuscript, we will be sure to clarify that our method will have potential scalability issues for very high-dimensional data ($d>5000$), e.g. genomics.

---

> ### Author Response · Authors · 2021-08-25
> **Additional comments**
>
> Thanks again for reviewing our paper.  Please let us know if there is anything else we can provide to help improve our score.  We believe that we have addressed the concerns mentioned in your review, and are happy to clarify further should you remain unsatisfied.
>
> Best, Authors of #4694

---

> ### Author Response · Authors · 2021-08-30
> **Additional comments 2**
>
> Given the short time left in this response period, we’d like to enquire whether our response addressed your concerns. If you found our response convincing, we kindly ask you to reconsider your score.  Thank you again for your time and consideration!

---

### Author Response · Authors · 2021-08-09
**General Response**

We wish to express our sincerest gratitude for taking the time to read and evaluate our manuscript.  We hope our responses satisfy the concerns raised. Should there still remain points that are unclear, please ask for additional clarification as we are happy to respond to additional inquiries.

---

### Author Response · Authors · 2021-08-21
**Additional comments**

We want to reiterate our sincerest gratitude to all the reviewers of our paper.  If you have any remaining concerns, please let us know - we would be happy to do our utmost to address them!

---

### Decision · Program_Chairs · 2021-09-27

**Decision:**

Accept (Poster)

**Comment:**

The reviewers have provided thoughtful and constructive comments. They have responded to the authors' feedback and the most active reviewers who are championing this manuscript continue to lean towards acceptance. I hope the authors will take the reviewers' comments to heart and encourage them to incorporate their thoughts in preparing the camera-ready version of their manuscript.